# Cell cycle networks link gene expression dysregulation, mutation, and brain maldevelopment in autistic toddlers

Tiziano Pramparo[1], Michael V Lombardo[2,3,4], Kathleen Campbell[1], Cynthia Carter Barnes[1], Steven Marinero[1], Stephanie Solso[1], Julia Young[1], Maisi Mayo[1], Anders Dale[1], Clelia Ahrens-Barbeau[1], Sarah S Murray[5,6], Linda Lopez[1], Nathan Lewis[7], Karen Pierce[1] & Eric Courchesne[1,*]

## Abstract

Genetic mechanisms underlying abnormal early neural development in toddlers with Autism Spectrum Disorder (ASD) remain uncertain due to the impossibility of direct brain gene expression measurement during critical periods of early development. Recent findings from a multi-tissue study demonstrated high expression of many of the same gene networks between blood and brain tissues, in particular with cell cycle functions. We explored relationships between blood gene expression and total brain volume (TBV) in 142 ASD and control male toddlers. In control toddlers, TBV variation significantly correlated with cell cycle and protein folding gene networks, potentially impacting neuron number and synapse development. In ASD toddlers, their correlations with brain size were lost as a result of considerable changes in network organization, while cell adhesion gene networks significantly correlated with TBV variation. Cell cycle networks detected in blood are highly preserved in the human brain and are upregulated during prenatal states of development. Overall, alterations were more pronounced in bigger brains. We identified 23 candidate genes for brain maldevelopment linked to 32 genes frequently mutated in ASD. The integrated network includes genes that are dysregulated in leukocyte and/or postmortem brain tissue of ASD subjects and belong to signaling pathways regulating cell cycle G1/S and G2/M phase transition. Finally, analyses of the CHD8 subnetwork and altered transcript levels from an independent study of CHD8 suppression further confirmed the central role of genes regulating neurogenesis and cell adhesion processes in ASD brain maldevelopment.

**Keywords** Autism Spectrum Disorder; brain development; co-expression; gene networks

**Subject Categories** Development & Differentiation; Network Biology; Neuroscience

**Mol Syst Biol (2015) 11: 841**

## Introduction

Autism Spectrum Disorder (ASD) is a heritable disorder involving early brain maldevelopment (Courchesne *et al*, 2011a). The brain at young ages is abnormal in a myriad of ways including brain overgrowth with an anterior/frontal to posterior cortical gradient in the majority, but undergrowth in a minority, during the first years of life (Courchesne *et al*, 2007); this shift upward in brain size distribution is quantitative and not categorical. Brain weight at autopsy is also shifted upward with heavier than the normal mean for an estimated 80% of 2–16 year olds, but lighter for a minority (Redcay & Courchesne, 2005; Courchesne *et al*, 2011b). A small sample of young ASD boys with heavy brain weight exhibited an excess of 67% neurons in the prefrontal cortex, which mediates social, communication and cognitive development (Courchesne *et al*, 2011b). The excess of neurons in enlarged brains points to potential dysregulation of mechanisms that govern cerebral cortical neuron number during second trimester development. Indeed, gene expression studies of prefrontal cortex in young ASD postmortem cases report dysregulation of gene expression associated with cell production, DNA-damage response, and apoptosis (Chow *et al*, 2012). Recently, alterations were detected in cell cycle timing and excess cell proliferation in neuroprogenitor cells derived from fibroblasts of living ASD patients who displayed early brain overgrowth (Marchetto *et al*, unpublished data).

1   Department of Neurosciences, UC San Diego Autism Center, School of Medicine University of California San Diego, La Jolla, CA, USA
2   Department of Psychology, University of Cyprus, Nicosia, Cyprus
3   Center for Applied Neuroscience, University of Cyprus, Nicosia, Cyprus
4   Autism Research Centre, Department of Psychiatry, University of Cambridge, Cambridge, UK
5   Scripps Genomic Medicine & The Scripps Translational Sciences Institute (STSI), La Jolla, CA, USA
6   Department of Pathology, University of California San Diego, La Jolla, CA, USA
7   Novo Nordisk Foundation Center for Biosustainability at the UCSD School of Medicine, and Department of Pediatrics, University of California San Diego, La Jolla, CA, USA
    *Corresponding author. Tel: +1 858 534 6914; E-mail: ecourchesne@ucsd.edu

Disrupting mechanisms regulating cell number in the second trimester has long been theorized to play a role in brain maldevelopment in ASD (Courchesne *et al*, 2001) because mutant mouse model studies show that cell cycle molecular machinery governs the overall size of the brain (Nakayama *et al*, 1996; Ferguson *et al*, 2002). In fact, several pathological changes characteristic of ASD were recently modeled in mouse (WDFY3 loss of function) and displayed abnormal decreases in cell cycle timing, excess radial glia cell proliferation, prenatal brain overgrowth, and an abnormal anterolateral to posteromedial gradient of cortical overgrowth; interestingly, it also displays focal laminar dysplasia associated with mis-migrated cells (Orosco *et al*, 2014). This latter pathology may also parallel the report of focal prefrontal and temporal cortical laminar defects in 91% of young ASD males and females (Stoner *et al*, 2014). However, additional genes are known to be associated with either abnormal brain enlargement or reduction in animal models (Ellegood *et al*, 2014) and/or rare individual ASD cases (O'Roak *et al*, 2012), suggesting additional mechanisms underlying ASD.

Recent genomic analyses of high-confidence genes in ASD (Parikshak *et al*, 2013; Willsey *et al*, 2013; De Rubeis *et al*, 2014) also point to dysregulation of cortical neuron number, laminar development, and cell cycle in the prefrontal cortex during second and third trimesters. While each of these genes occurs only in rare individual ASD cases, cycle cell dysregulation functions may commonly disrupt development in the second trimester in ASD. While many high-confidence ASD genes regulate downstream transcriptional programs including cell cycle functions and proliferation, such as CHD8, in general they are not cell cycle genes per se. This suggests that effects of high-confidence ASD genes on cell proliferation and brain size may be quantitative and continuous and not categorical. Many genetic and non-genetic defects could disrupt cell cycle, with changes in signaling and transcriptional activity, which could lead to variations in cell number and brain size. Unfortunately, it remains infeasible to directly test the impact of cell cycle changes on cell number and brain size in ASD *in vivo* or with postmortem approaches. This is because cell cycle activity changes with development, and assays that test cell cycle activity in older postmortem tissue provide only indirect information about its function during fetal development. Moreover, the scarcity of postmortem ASD cases further limit getting even indirect evidence of cell cycle dysfunction on brain size from this avenue.

These barriers hinder the study of relationships between cell cycle disorganization and brain size variance in ASD during early development. However, we note that genetic disruption of cell cycle network organization could be detectable in multiple tissues at different ages. While the physiological response of a genetic perturbation often varies with tissue type and age, the presence of disruption may nonetheless be detectable and quantifiable across types and ages. That is, detection in one tissue type at one age, such as leukocytes in infants, may help the search for the presence of disruption in other inaccessible cell types and ages, such as fetal neuroprogenitor cells. Of note, the GTEx Consortium reported in Science that cell cycle gene expression networks are present in all tissues, including brain and blood (GTEx Consortium, 2015).

Therefore, we took a systems biology approach to analyzing gene co-expression patterns in blood leukocyte samples of ASD and control infants and toddlers in order to examine how variation in co-expression modules are associated with variation in brain size at very young ages in ASD. Here, we show that gene expression profiles from leukocytes at very young ages may be a biomarker of early brain growth deviance in ASD. Furthermore, we use cell cycle networks as an entry point to elucidate perturbation of transcriptional networks associated with smaller and bigger brains. Our findings of network dysfunction are integrated with recent genomic studies describing genes frequently mutated in ASD, thus providing compelling evidence that cell cycle networks may indeed be a point of convergence for gene expression dysregulation, mutation, and early brain maldevelopment in ASD.

## Results

We tested the hypothesis that blood-based gene expression profiling may reveal biological signatures relevant to neurodevelopment and that such signatures may differ between ASD and control toddlers. Leukocyte RNA levels were analyzed in relationship to total brain volume (TBV) using an established approach based on gene co-expression (WGCNA; Fig 1A and B; Langfelder & Horvath, 2008). This method elucidates patterns of altered gene expression, organized as networks of co-expressed genes, and provides insights into relationships of genes with disease-related endophenotypes or traits. Furthermore, it provides metrics to understand the details of network perturbation (Langfelder & Horvath, 2007, 2008; Fig 1C). We leveraged network metrics to understand whether network perturbation differentially affected smaller and bigger brains in ASD toddlers as compared to controls (Fig 1D). Lastly, we used a reverse genetic approach to frame our findings with recent evidence from genomic studies reporting high-confidence genes of ASD (De Rubeis *et al*, 2014).

### Different gene networks associate with brain size in ASD and control toddlers

Analyses were run using processed gene expression data (Pramparo *et al*, 2015) that included 12,208 unique gene-probes from 87 ASD and 55 control male subjects ages 1–4 years. The majority of subjects were of Caucasian origin and Pearson's chi-squared test showed no significant difference in race characteristics between ASD and control ($X^2_{[5]} = 7.98$, $P = 0.1569$). Multivariate regression analysis showed no variance explained by differences in race and ethnicity characteristics between ASD and control subjects and age variance was accounted in downstream analyses. Unsupervised WGCNA resulted in 22 co-expression modules (Appendix Fig S1). Each module was given an arbitrary color name and was summarized by a metric known as the module eigengene (ME), which is the first principal component of the module (i.e., axis capturing the majority of variation in expression in the module). After degree-preserving random shuffling, it was determined that all 22 modules were significantly detected above chance levels (see Materials and Methods and Appendix Fig S2). Module preservation analysis between the two separate datasets (ASD/control) displayed high-preservation scores, suggesting that the combined analysis was not confounded by differences in networks structure of the two datasets (Appendix Fig S3 and Appendix Table S1).

Module eigengene values from each of the 65 ASD and 38 control subjects, who also had MRI scans, were used in linear correlation

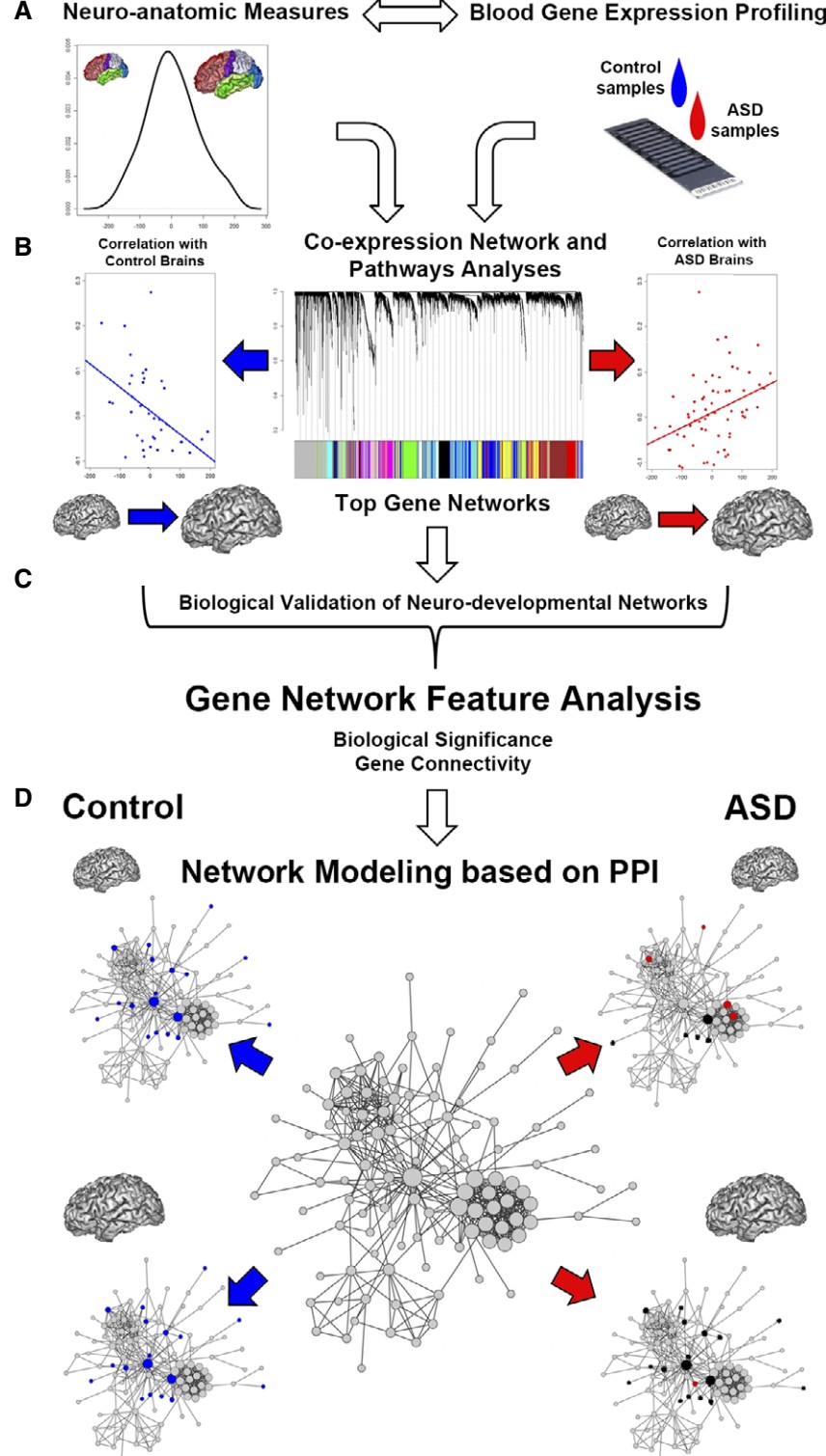

**Figure 1. Schematic of the approach used in the study.**

A  Blood gene expression was analyzed in relationship with neuroanatomic measures using a co-expression network-based approach (WGCNA). The distribution of neuroanatomic measure was normal and not significantly different between ASD and control toddlers. The analysis of co-expression was combined with all available samples.

B  Data from the combined network-based analysis was further investigated in each ASD and control group separately using a linear model.

C  Network features, calculated from the WGCNA co-expression analysis in relationship to brain size, were used to dissect alterations of network patterns in ASD brains.

D  Network features were also used to characterize smaller and bigger brains in each study group.

tests in which we related RNA levels to brain size. TBV measures were age-corrected (see Materials and Methods) and showed normal distributions with no statistically significant differences in mean and variance between the two groups (Fig 2A). Seven modules were significantly correlated with TBV measures across all subjects (FDR < 0.05) with the greenyellow and grey60 gene modules displaying the strongest correlations (Fig 2B). Permutation analysis with randomly generated MEs (see Materials and Methods) demonstrated that these associations were significant against chance for all but the yellow module (Appendix Fig S4).

To identify gene networks that correlated with brain size within each diagnostic group, we computed Pearson's $r$ correlation statistics between each of the seven MEs and TBV measures in ASD and control toddlers, separately. In control toddlers, the greenyellow

and grey60 MEs were significantly correlated with age-corrected TBV (Fig 2B), while brain size in ASD toddlers displayed significant linear correlations with the salmon, turquoise, and cyan MEs (Fig 2B; see Appendix Table S2 for bootstrapped 95% confidence intervals). We restricted further analyses to only modules showing the strongest effects on brain size (i.e. $r > 0.3$, $P < 0.05$, FDR < 0.05), which included greenyellow, grey60, and salmon modules. These effects were found to be independent of age (Appendix Table S3) and confirmed to be significant against chance after permutation analysis (Appendix Fig S5 and see Materials and Methods). WGCNA on the separate ASD and control datasets also confirmed that these three gene networks were the strongest signal associated with TBV variation in each group (Appendix Figs S6 and S7 and Dataset EV1). We next used permutation tests to examine

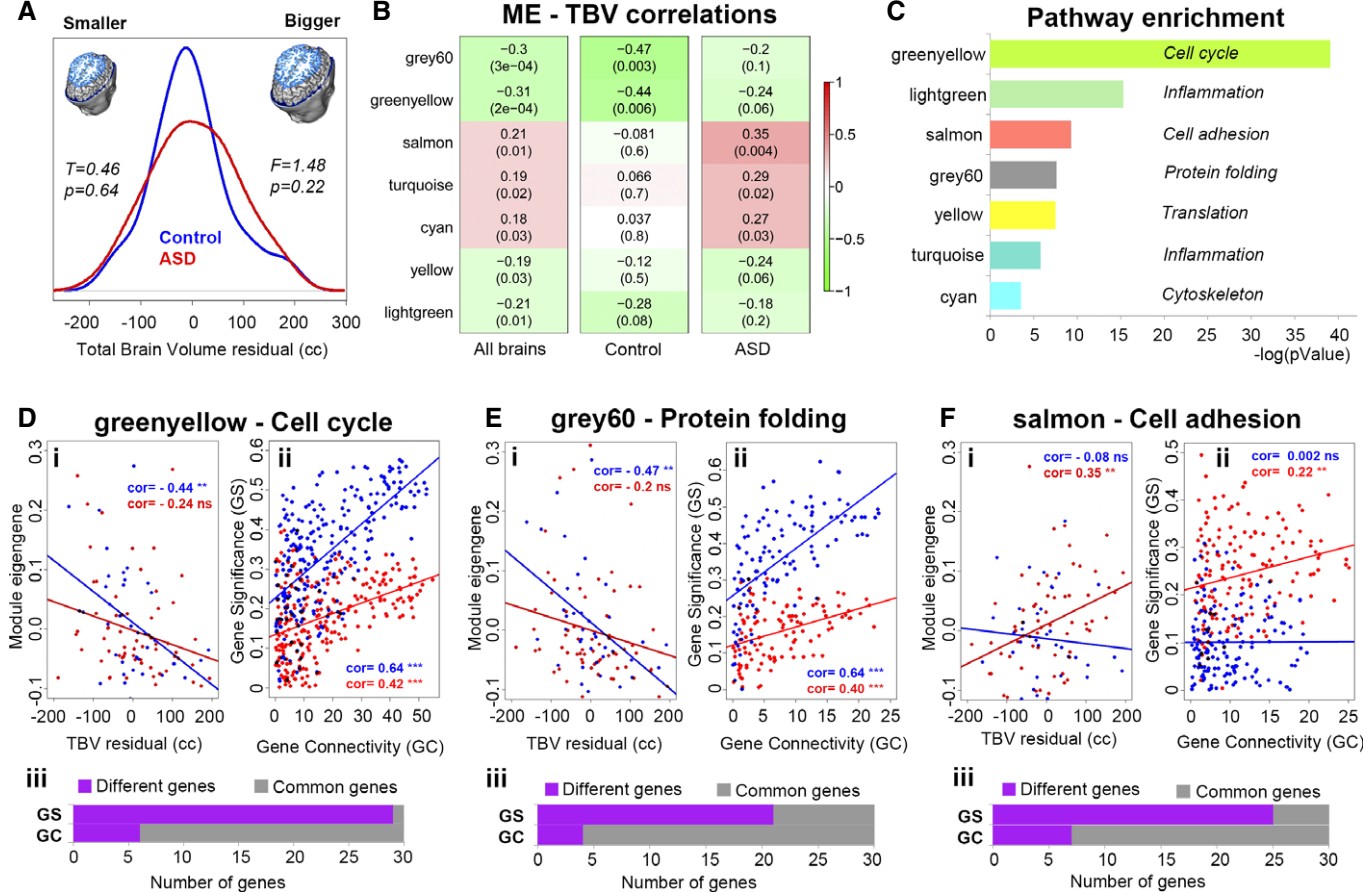

**Figure 2. Analysis of gene networks associated with variation in brain size in ASD and control toddlers.**

A    Distributions of brain size as indexed by total brain volume (TBV) in ASD and control toddlers used in the co-expression analysis (WGCNA). T, value from *t*-test; F, value from Levene's test.

B    Module eigengenes (MEs) from the combined WGCNA are linearly correlated with TBV measures in all brains, ASD and control groups. *P*-value is in parenthesis and adjusted *P*-value (*q*-value) is < 0.05 for all seven modules. Significant associations after 10,000 permutation tests are provided in Appendix Figs S4 and S5.

C    Metacore enrichment scores of the seven (7) modules initially related to brain size variation across all subjects. Each module is called by its assigned color and represents the top process network obtained by the enrichment analysis in Metacore GeneGO (see also Dataset EV1).

D–F  (i) Linear modeling of module eigengenes (MEs) by TBV measures in control (blue) and ASD (red) toddlers. See also Fig 2B for cor and *P*-values. (ii) Linear modeling of GS by GC to display changes in network organization relevant to brain size. (iii) The top 30 genes with highest values for GS and GC were compared between ASD and control. Purple indicates the number of genes that moved away from the top 30 rank position between the two groups (Different genes), and grey indicates the number of genes that did not (Common genes). Significance codes: ****P*-value < 0.001; ***P*-value < 0.01; cor, correlation coefficient; ns, not significant.

whether the strength of these correlations (MEs-TBV) significantly differed between the two groups. However, correlations were not significantly different between groups for greenyellow $P = 0.33$, but were at or beyond trend level significance for grey60 ($P = 0.06$) and salmon ($P = 0.01$; see Fig 2Di–Fi and Appendix Table S4).

To investigate the biological functions of these modules, we ran pathway enrichment analysis in Metacore GeneGO using a threshold of FDR $q < 0.05$. The greenyellow and grey60 modules were enriched in genes with cell cycle and protein folding functions, respectively, while genes in the salmon module were enriched in cell adhesion functions (Fig 2C and Dataset EV1). This enrichment remained significant after filtering for expression in fetal and adult brain tissue using the Metacore GeneGo database (Dataset EV1). The other modules with modest correlations displayed enrichment in translation, inflammation, and cytoskeleton rearrangement functions (Fig 2C and Dataset EV1).

## Network perturbation in ASD affects gene connectivity and relevance for brain size

In addition to quantifying gene module summary measures like the module eigengene and its relationship to brain size, we also used two gene-level metrics (gene significance and gene connectivity) to assess associations with brain size. Gene significance (GS) is defined as the correlation between gene expression and a trait (i.e., TBV), thus providing a measure of "significance or relevance" of a particular gene to variation in a trait such as TBV. Gene connectivity (GC) is a connectivity measure indicating how strongly connected (i.e., correlation strength) is a particular gene with all other genes within the module. Higher GC values are indicative of central or 'hub' genes, whereas genes with lower GC are oriented around the periphery of the co-expression module. Examining the correlation between GC with GS values for each gene allows for insight into understanding how metrics of a gene's organization within a network (i.e. gene connectivity) may be associated to its relevance with brain size (i.e., gene significance).

GS-GC correlations were stronger in control compared to ASD (i.e. more positive) in both the cell cycle (control: $r = 0.64$; ASD: $r = 0.42$; $z = 3.47$) and protein folding modules, (control: $r = 0.64$; ASD $r = 0.40$; $z = 3.19$) (see Fig 2Dii–Fii). Thus, as a gene becomes more highly connected with other genes within the cell cycle and protein folding modules, it also becomes more relevant to (or has stronger impact on) TBV, and this relationship is stronger in control than ASD. For the cell adhesion module, the ASD group showed a stronger correlation between GS and GC than the control group (ASD: $r = 0.22$; control: $r = 0.002$; $z = 2.48$; see Fig 2Dii–Fii). Thus, as a gene becomes more highly connected within the cell adhesion module, it becomes more relevant to brain size in ASD than in the control group. Along with the evidence showing generalized atypicality in GS in ASD (i.e., reductions in GS in cell cycle and protein folding modules, but increase in cell adhesion; Appendix Fig S8), this evidence supports the idea that GS is accompanied by a modest alteration in GC between groups, indicating that a gene's relevance to brain size covaries with changes in network organization in ASD (Fig 2Dii–Fii). With regard to the cell cycle network in particular, this network re-organization in ASD can be described as many high GC genes (i.e., hub-genes located more centrally within the network) with a reduced GS, but also many low GC genes (i.e.

low-connectivity genes located around the periphery of the network) which displayed some of the highest GS levels (Fig 2Dii–Fii).

Of the three modules, the cell cycle module displayed the most severe network re-organization. This can be shown through further analyses of the top 30 genes on each metric (GS and GC; see Materials and Methods). First, we ran Venn analyses to determine the gene overlap between ASD and control toddlers and found that the majority of the genes with highest GS were unique to each group, especially for the cell cycle gene network with 29 out of 30 genes being different between groups (Fig 2Diii and Dataset EV1). Then we investigated whether the top 30 GC genes (i.e., hub-genes) in the co-expression network were also the top GS genes in each group. Within the cell cycle module, 16 of the 30 hub-genes also possessed the top GS scores for controls, while in ASD only 5 of the 30 hub-genes were top GS genes (OR = 5.71, $P = 0.004$ CI = 1.72–18.94). The remaining 25 top GS genes in ASD had lower GC scores, and thus were considered 'peripheral' in the cell cycle co-expression network. While in controls these 25 peripheral genes displayed a strong positive association between GS and GC ($r = 0.74$, $P = 2.7e\text{-}5$), in ASD the directionality of the association flipped ($r = -0.34$, $P = 0.098$; Appendix Fig S9), resulting in a substantial group difference in correlation strength ($z = 4.33$, $P = 1.51e\text{-}5$). For the protein folding module, there were similar proportions of hub-genes displaying top GS scores in both groups (OR = 1.96, $P = 0.198$, CI = 0.70–5.48). However, among the peripheral genes (i.e. top GS genes with low GC scores), again there was a flip in directionality of GS-GC correlation (controls $r = 0.89$, $P = 1.8\text{-}e\text{-}6$; ASD $r = -0.19$, $P = 0.45$; $z = 0.35$, $P = 8.60e\text{-}8$; Appendix Fig S9). Likewise, the cell adhesion module also displayed similar proportions of hub-genes with top GS scores (OR = 1, $P = 1$, CI = 0.30–3.30). Peripheral genes (i.e., top GS genes with low GC scores) display similar GS-GC correlations across controls and ASD (controls $r = 0.04$, $P = 0.85$; ASD $r = -0.35$, $P = 0.094$; $z = 1.34$, $P = 0.17$ Appendix Fig S9).

This evidence reinforced the findings of network re-organization and revealed a trend in gene expression relevance for brain size that shifts from central genes (hub-genes) in control toddlers to peripheral genes in ASD toddlers particularly within the cell cycle network. The overall network perturbation may underlie potential downstream consequences in overall transcriptional regulation.

## Cell cycle module is preserved and highly expressed during early stages of normal fetal brain development

We next reasoned that if correct neurodevelopment relies on the tight modulation of gene networks driving brain size, changes in expression levels of these networks would likely be most damaging at early developmental stages. It is also expected that biological processes involved in cell proliferation (e.g., during the neuronal progenitor pool expansion) would be expressed at high levels at earliest ages and lowest during postnatal life when the brain structures have already been formed.

Based on these hypotheses, we utilized gene expression data from the Allen Institute BrainSpan Atlas (Miller *et al*, 2014), in which control human brain tissues were expression-profiled at multiple time points from early prenatal stages to adulthood. First, we examined which of the modules detected in blood would be highly preserved in a WGCNA on BrainSpan data. Several of the 22

blood-modules were moderately to highly preserved (i.e., moderate = Zsummary values between 2 and 10; high = Zsummary > 10; Fig 3A; see statistics in Appendix Table S5). The cell cycle module (greenyellow) showed the highest level of preservation (e.g., Zsummary = 14 and top median rank), suggesting that its network structure is highly preserved in human brain samples taken at different developmental stages. Due to its relevance for development, we next focused on the cell cycle module and investigated the developmental trajectory of the gene expression and specifically tested the hypothesis that expression would be upregulated in prenatal vs postnatal time points. Indeed, BrainSpan cell cycle ME values from prenatal time points were much higher than at postnatal time points (Wilcoxon rank sum $z = 3.51$, $P = 4.44e-4$; Fig 3B and C). Of note, consistent with prior literature indicating that cell cycle processes are important to drive early brain development, our findings provide

postnatal *in vivo* evidence of gene expression alterations in ASD that can be traced back to early developmental stages and that are relevant to brain size.

**Network perturbation is more pronounced in bigger ASD brains**

We next wanted to test whether the cell cycle module showed high levels of protein–protein interactions (PPIs) and how perturbation of these PPI genes would be relevant to brain size variation in ASD. This analysis would also provide independent validation that gene products (proteins) within the same biological process, such as cell cycle, display consistent perturbations; we have detected from the co-expression analysis. In addition, we designed the PPI analysis to determine whether smaller brains significantly differed from bigger brains in ASD compared to controls.

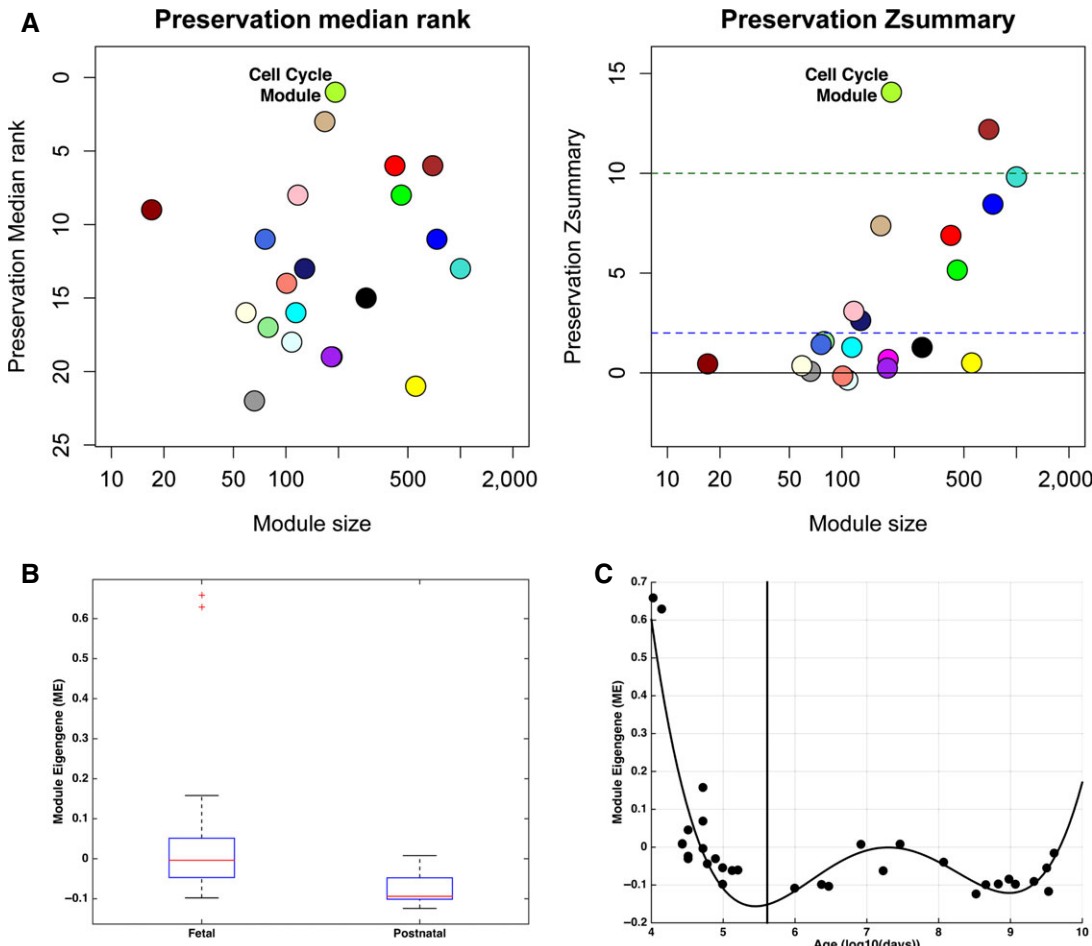

**Figure 3.  BrainSpan preservation analysis and cell cycle developmental trajectory.**

A    Preservation analysis between BrainSpan dorsolateral prefrontal cortex gene expression data and the ASD + control blood data. Zsummary statistic (e.g., Zsummary > 10 means highly preserved, Zsummary in between 2 and 10 is weak to moderate preservation, Zsummary < 2 is little to no preservation and median rank (modules with lowest rank are highly preserved)). Median Rank and Zsummary values indicate high module preservation between the two datasets.

B    Boxplot showing module eigengene (ME) values the BrainSpan cell cycle module for fetal versus postnatal time points (15 fetal versus 16 postnatal time points). The box refers to the interquartile range (IQR), which we refer to as Q1 (25th percentile) and Q3 (75th percentile).  The upper whisker represents Q3 + 1.5*IQR, while the lower whisker represents Q1 − 1.5*IQR.  The line in the middle of the box represents the median.

C    Scatterplot indicating the BrainSpan cell cycle module trajectory across development (vertical line indicates birth; time points to the left of the line are fetal time points, while time points to the right of the line are postnatal time points; best-fit curve indicates a 4th order polynomial fit).

A cell cycle reference network based on PPI was constructed by querying all 253 genes from the cell cycle module using the DAPPLE database (Rossin *et al*, 2011). Out of these genes, 119 displayed a higher number of PPIs compared to chance ($P < 0.001$; Appendix Fig S10; see PPI stats in Dataset EV1). We then mapped onto this reference PPI network the top 30 hub-genes relevant to brain size that we previously identified in the combined co-expression network analysis (Fig 2Diii–Fiii). Of the top 30 hub-genes, 19 and 20 genes (in control and ASD, respectively) mapped into the PPI network (Dataset EV1). For each of these, 19 and 20 hub-genes in the PPI network, we re-calculated their GS scores and *P*-values (see Materials and Methods) for subgroups with either smaller (GS-SM, residuals < 0 cc) or bigger (GS-BG, residuals > 0 cc) than the mean average brain size (see Fig 2A) and compared their GS scores

with the GS scores calculated from the previous combined analysis using all brains (GS-ALL). We used the lowest GS-ALL value as threshold in each group and observed whether the new GS-SM or GS-BG values passed this GS-ALL threshold. When a gene did not meet the GS-ALL threshold in GS-SM or GS-BG subgroups, we considered the gene as losing its relevance to brain size within the subgroup. If a gene passed the GS-ALL threshold in GS-SM or GS-BG subgroups, we considered that gene as retaining or gaining relevance within the subgroup.

Surprisingly in controls, all 19 genes passed the GS-ALL threshold in both the small and big brain subgroups (Fig 4A and Dataset EV1). This suggested that brain growth in control toddlers is possibly driven by a common set of genes regardless of size variation. In ASD, 14 of these 19 genes passed the GS-ALL threshold in

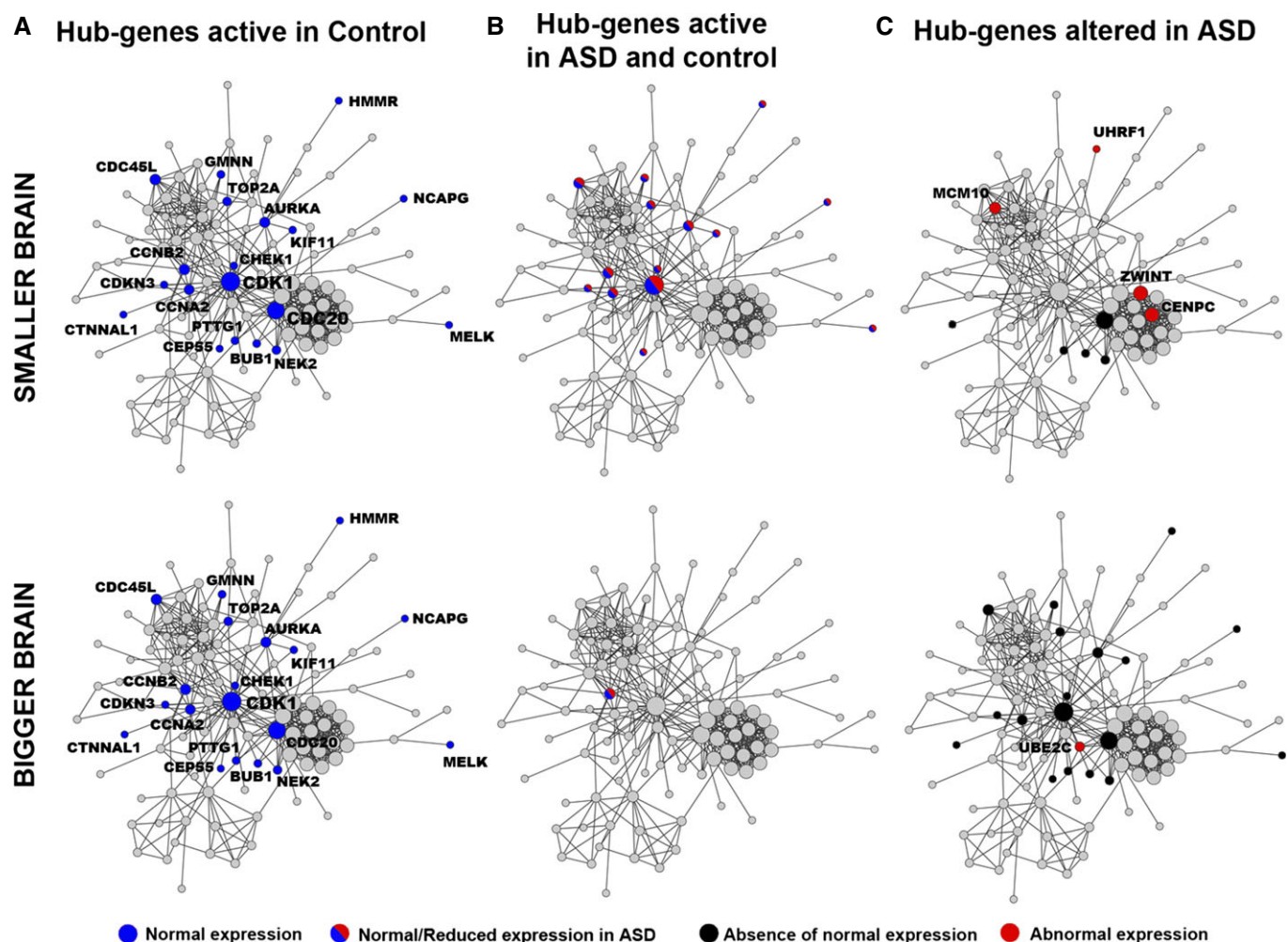

**Figure 4. Topological analysis of cell cycle hub-genes in a protein–protein interaction (PPI) network.**

Mapping of the top 30 hub-genes relevant to TBV measures in smaller and bigger ASD and control brains.

A  Hub-genes that are normally active in control toddlers.
B  Hub-genes that are active in both control and ASD toddlers.
C  Hub-genes that are abnormally active in ASD toddlers.

Data information: Node size (circle) is proportional to the number of actual biologically driven PPIs. Blue and red nodes are hub-genes that displayed PPIs and were in the top 30 lists in control and ASD, respectively. Grey nodes are genes in the cell cycle PPIs network that were not ranked within the top 30 genes. Black nodes are genes that fall out from the top 30-gene list in ASD compared to control.

the small brain subgroup (Fig 4B and Dataset EV1). The difference in these ratios (i.e. 19/19 in controls vs 14/19 in ASD) was significant (OR = 14.8, $P$ = 0.02, CI = 0.75–289.4). Conversely, in ASD only 1 gene (*CCNB2*) passed the GS-ALL threshold in the big brain subgroup (Fig 4B) and this difference in ratios (i.e. 19/19 in controls vs 1/19 in ASD) was highly significant (OR = 481, $P$ = 3.35e-9, CI = 18.4–12,570). Lastly, we identified the genes that were unique to smaller or bigger ASD brains, thus either lost or gained relevance for size. A total of 9 hub-genes were found abnormally correlated (*ZWINT, CENPE, MCM10,* and *UHRF1*), or uncorrelated (*CDC20, BUB1, NEK2, PTTG1,* and *CCNE2*) with smaller ASD brains. Bigger ASD brains instead displayed greater PPI network alteration in respect to smaller brains with 18 genes (Fig 4C and Dataset EV1) losing correlation strength for size, while 1 gene (*UBE2C*) displayed a gain in correlation for bigger brains (14 and 9 in smaller vs 1 and 18 in bigger ASD brains; OR = 28, $P$ = 2.2-e-4, CI = 3.2–248; Fig 4C).

In total, we identified 23 co-expressed hub-genes within the cell cycle PPI network abnormally related to brain size variation in ASD that are good candidate genes for brain maldevelopment (Dataset EV1). These findings added substantial biological validation to the above genetic interaction analyses that showed genes governing brain size in ASD and controls are substantially different from each other. Furthermore, while such differences are present for both smaller and bigger ASD brains, the difference from controls is much greater for bigger ASD brains compared to smaller ASD brains.

## Cell cycle networks link gene expression dysregulation, mutation and brain maldevelopment

A large body of evidence suggests that developmental alteration of genes that control cell number may underlie neuropathology of ASD (Courchesne *et al*, 2001, 2007, 2011b). Moreover, recent genomic studies identified High-confidence (Hc) genes with loss-of-function mutations that are involved in large networks including functions in the regulation of cortical cell number production (Willsey *et al*, 2013; De Rubeis *et al*, 2014). Two Hc genes were also found mutated in subjects with significantly smaller (DYRK1A) and bigger (CHD8) heads (O'Roak *et al*, 2012).

To identify specific molecular mechanisms of dysregulation, we tested the hypothesis that these Hc genes may be at least partially involved in the upstream regulation of the 23 PPI candidate genes we identified abnormally expressed in ASD. Thus, we constructed an Hc network by using 32 Hc genes (out of the 33 reported in De Rubeis *et al*, 2014) in Metacore GeneGO to identify their direct downstream targets. These Hc genes are associated with pleiotropic roles including not only the regulation of synaptic processes, but also the regulation of cell number and expression of other downstream genes (Dataset EV1) (Geschwind & State, 2015). The Hc network included a total of 414 genes (Fig 5). Similarly, we queried the same database to obtain the list of genes with regulatory functions upstream of the 23 PPI candidate genes and mapped these genes in the Hc network. One hundred and six (106) genes were identified as upstream regulatory. A schematic description of the network construction can be found in Fig EV1.

Analysis of the integrated Hc network demonstrated that four of the 23 PPI candidate genes (*AURKA, CDK1, CCNA2* and *CCNE*) were direct downstream targets of Hc genes (*CHD8, ARID1A/B,* and *CUL3*) and upstream regulators of other PPI candidate genes (Fig 5). Using

differentially expressed (DE) from a companion study (Pramparo *et al*, 2015), we found that eight Hc genes (*CHD8, ARID1A, ASH2L, ACTB, NR3C2, SUV420H1, ADPN,* and *MYO9B*) were DE in leukocytes obtained from ASD and control (Fig 5). Four other Hc genes (*CUL3, SYNGAP1, NAA15,* and *ARID1B*) were upstream regulators of the PPI candidate genes and *CUL3, SYNGAP1,* and *NAA15* were also DE in postmortem brain tissue (Voineagu *et al*, 2011; Pramparo *et al*, 2015; Fig 5). Several other upstream regulatory genes were included in the Hc network, such as *E2F1, MYC, GSK3, YWHAB, ESR1, EGFR, PCNA, CDKN1,* and *ERBB1*. Most importantly, many key upstream regulatory genes were DE in leukocytes, such as *CHD8, ARID1A, AKT1,* Beta-catenin (*CTNNB1*), *SMAD3, CREB1,* and *NOTCH1* and/or in postmortem brain tissue (*TCF4, CREB1, SMAD3, CAMK2A, LIMK1, NCOA3, CCNE1,* and *BRD2*).

We found an over-representation of DE genes, previously identified in (Pramparo *et al*, 2015), among the upstream regulatory genes ($n$ = 106) in the Hc network ($n$ = 414) (see Materials and Methods; *Hyp. P* = 1.e-12), suggesting their role in regulating the cell cycle network. Further pathway analysis of the upstream regulatory genes displayed a strong enrichment in processes involved in the regulation of cell cycle phase transition (G2-M and G1-S) under the master regulation of *APC* (inducing progression and exit from mitosis), *ESR1* (promoting G1-S transition) and *ATM* (initiating G2-M arrest) genes (Dataset EV1). Top signal transduction pathways were PTEN ($P$ = 2e-16, FDR = 4e-14), ESR1 ($P$ = 5.7e-14, FDR = 2.1e-12) and NOTCH ($P$ = 2.2e-13, FDR = 4.4e-12). These findings demonstrate a clear connection between the dysregulation of cell cycle gene networks and abnormal brain size/development in ASD toddlers. Overall, a large proportion of genes in the Hc network were DE in postmortem brain tissue (*Hyp. P* = 2.6e-11), demonstrating the validity and relevance of the Hc network in brain tissue development and function. Lastly, a significant enrichment was found also for SFARI ASD candidate genes (excluding the 32 Hc query genes; 706 curated SFARI genes Updated March, 2015) within the Hc network (*Hyp. P* = 1.4e-13), further reinforcing the link of the cell cycle network to ASD.

## CHD8-subnetwork perturbation causes significant loss of association with brain size in ASD via E2F-dependent genes expressed at G1-S phase transition

We further analyzed the Hc network to pinpoint which of the Hc subnetworks displayed gene expression levels associated with normal brain size and whether the dysregulation of gene expression in ASD would significantly disrupt this association with ASD brains (see Materials and Methods). Hc subnetworks were defined based on each Hc gene as central hub and all the genes that directly linked to it. We then extracted expression values from our dataset for all genes in each Hc subnetwork and calculated the average expression value for each subnetwork. For instance, CHD8 was directly linked to a total of 25 genes (Fig 6A) and 16 had expression values; thus, the average expression for the CHD8 subnetwork was calculated using these 16 genes. Next, we ran linear correlations between each average subnetwork expression value and TBV measures and tested whether the strength of these correlations significantly differed between ASD and control groups. Out of all the Hc subnetworks, three were found with significant associations (CHD8, NR3C2, APH1A), and only the CHD8 subnetwork displayed significant group

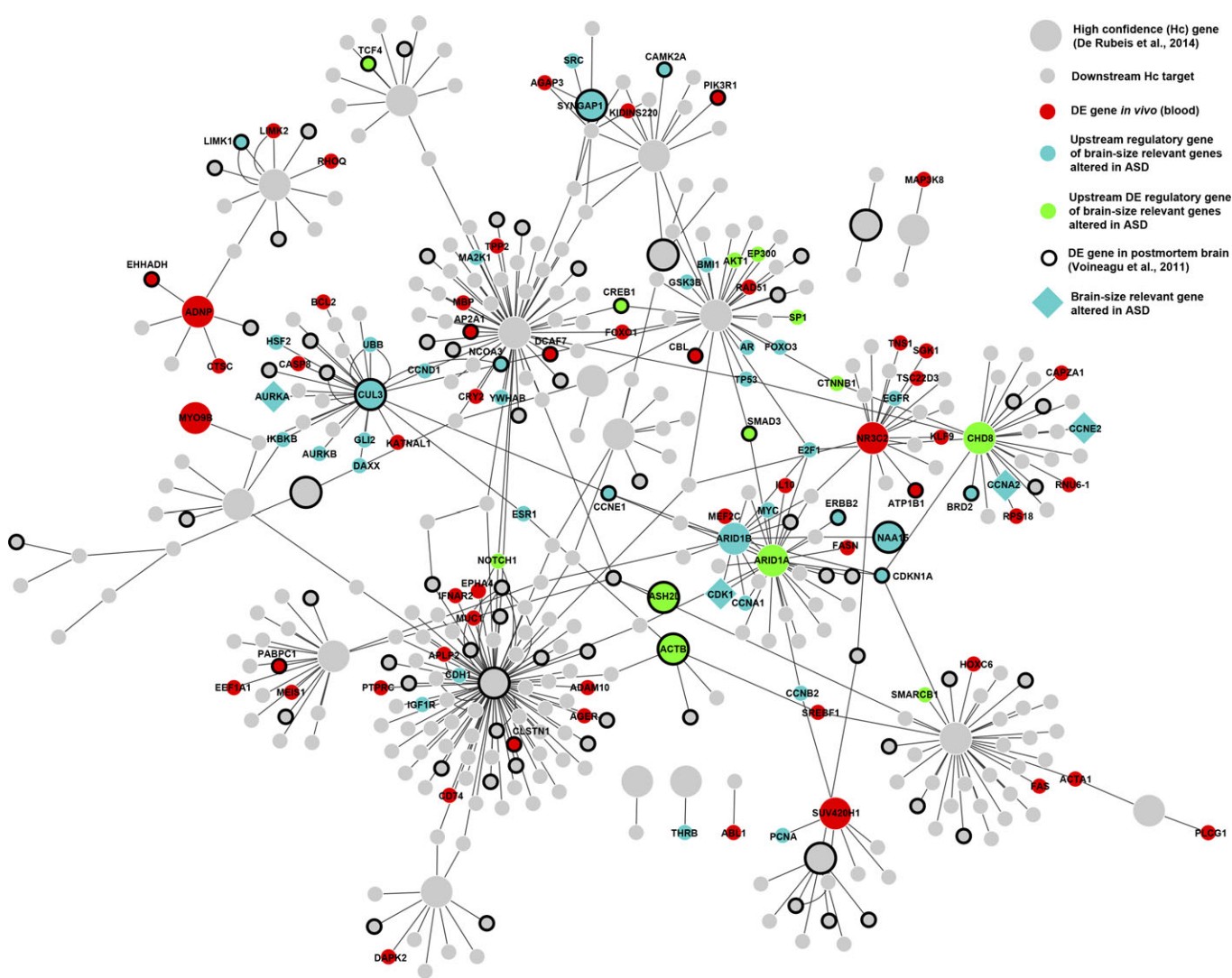

**Figure 5. High-confidence (Hc) network.**
A Hc network was generated following the approach represented in Fig EV1. Big nodes represent Hc genes (grey or colored circles). Small grey or colored circles represent direct downstream targets of the Hc genes. Colors code for different categories of genes mapped into the Hc network. Red indicates genes that are differentially expressed (DE) in blood of the same subjects described in this study. Cyan indicates genes that are upstream and regulate any of the 23 brain-relevant genes identified in this study. Green indicates genes that are both DE and regulators of the 23 brain-relevant genes. Big cyan diamond shapes are brain-relevant genes that mapped into the Hc network. Bold circle outlines represent genes that are DE in postmortem brain tissue of ASD donors.

differences via permutation analysis (see Appendix Table S6, Fig 6B). The significant association of the CHD8 subnetwork in controls was found to be independent of age by linear modeling and ANOVA testing (see Appendix Table S7).

This evidence suggested an important role of CHD8 on brain size in ASD which is strongly supported by a growing literature describing its mutations (O'Roak *et al*, 2012; Bernier *et al*, 2014; McCarthy *et al*, 2014; Prontera *et al*, 2014) and function (Rodriguez-Paredes *et al*, 2009; Subtil-Rodriguez *et al*, 2014; Sugathan *et al*, 2014).

Recent *in vitro* studies aiming to define the cellular phenotypes and downstream targets of CHD8 supported the relevance of this subnetwork to brain size. Indeed, five brain-relevant genes included in the CHD8-subnetwork (*CCNE2, TYMS, CCNA2, CDC6,* and *BRCA2*) were reported with impaired expression upon CHD8 knockdown

(Rodriguez-Paredes *et al*, 2009; Subtil-Rodriguez *et al*, 2014). Dysregulation was found specific for E2F-dependent genes expressed during G1-S phase transition in proliferating cells, supporting our findings that point to this mechanism as underlying brain maldevelopment in ASD (Subtil-Rodriguez *et al*, 2014). To test whether potential downstream effects of CHD8 would converge to the same gene networks we found dysregulated in ASD, we ran pathway analysis using the CHD8-targets identified by ChIP-chip in two independent studies (Sugathan *et al*, 2014; Cotney *et al*, 2015). Most importantly, we ran enrichment on the common targets from these studies. Analysis of the common genes found DE and CHD8 bound in neuronal progenitor cells (NPCs) from the Sugathan study (Sugathan *et al*, 2014) showed enrichment in development, signaling processes, and transcription as well as DNA-damage and cell cycle functions (Dataset

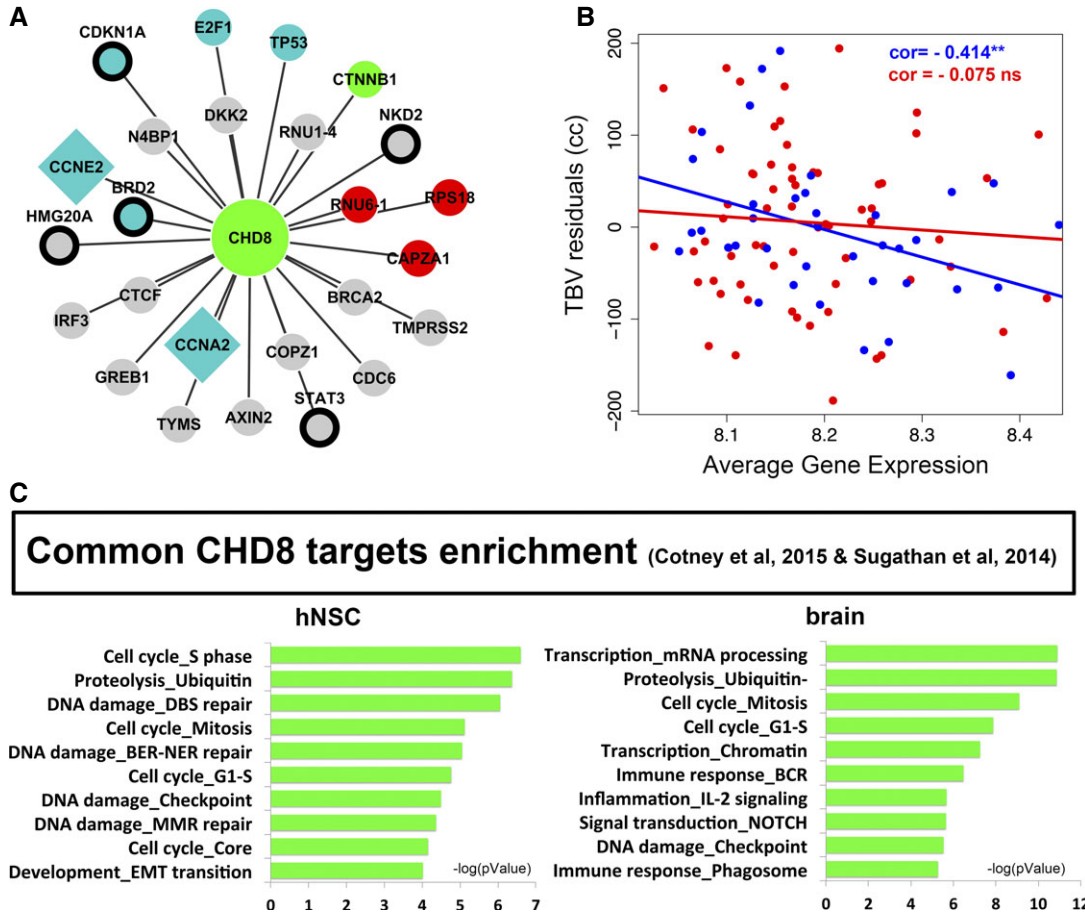

**Figure 6. CHD8 subnetwork analysis in relationship to brain size and downstream CHD8-knockdown effects *in vitro*.**

A CHD8 subnetwork analysis included all CHD8-targets from the Hc network. Network legend is the same as in Fig 5: grey (downstream CHD8 target), cyan (upstream regulatory gene of brain-size relevant genes altered in ASD), red (differentially expressed in blood), green (cyan and red), diamond shape (brain-size relevant gene altered in ASD), and black circle (differentially expressed in ASD cortex from Voineagu et al, 2011).

B Linear correlation analysis of gene expression levels with TBV measures in ASD and control toddlers. See permutation analysis in Appendix Table S6. **P < 0.001.

C Pathway analysis in Metacore. FDR < 0.05.

EV1). Common targets from Sugathan *et al* and Cotney *et al* (hNSC and brain genes, $N = 1,620$ and $N = 1,662$, respectively) studies showed a strong enrichment for cell cycle and DNA-damage processes with prominent S phase functions (Fig 6C). Of note, we looked for enrichment in perturbed cell cycle genes that we identified (greenyellow module) using the common targets from Sugathan *et al* and Cotney *et al* (hNSC and brain genes) and we found significant enrichment when using the hNSC genes (*Hyp. P* = 6.3e-5).

Altogether, these data suggest a role for CHD8 in development. Upon disruption, its downstream effects converge to genes and networks we have found predicting risk for ASD, associated with brain maldevelopment in ASD and pointing to mechanisms that affect regulation of cell phase transition during mitotic cycle via E2F-dependent genes.

## Discussion

This study of *in vivo* gene expression relationships with early brain size addresses questions relevant to ASD patho-etiology and

physiology. It is currently unknown whether blood gene expression can be used to identify biomarkers of brain maldevelopment *in vivo* at early ages and/or whether these biomarkers could elucidate the molecular mechanisms underlying fetal cortical maldevelopment. Several lines of evidence suggest that aberrant regulation of cell number may underlie the ASD neuropathology during the second/third trimester of prenatal life. However, fundamental limitations, related to the inaccessibility of the tissue *in vivo* and the paucity of young postmortem samples, prevent the testing of this hypothesis with both exploratory and targeted research. Our study represents a step in this direction and aims to lay the groundwork for further hypothesis-driven investigations. A systems-level analysis of biological networks is indeed required to build a framework to understand the spatio-temporal scale, the effects of perturbation and the resulting physiological states characterizing diseases (Ge *et al*, 2003; Somvanshi & Venkatesh, 2014). Thus, our findings suggest a point of reference to further the study of alterations in genes and biological processes that underlie neuropathology of ASD.

Here, we identified gene networks—cell cycle and protein folding —in blood that strongly correlate with early brain size in control

toddlers. We additionally identified dysregulated gene networks in blood that correlate with early brain maldevelopment in infants and toddlers with ASD. These gene networks retained functional enrichment in human brain tissue and displayed consistent expression profiles during cortical fetal development.

The combined findings of this study suggest that there is a prenatal disruption of neuron number regulation, cell differentiation, and overall architecture of the developing cortex in ASD. Underlying this disruption is an abnormal functional organization of cell cycle and protein folding gene networks and the abnormal activation of other functional networks, such as cell adhesion. Thus, our evidence suggests that brain maldevelopment in autistic infants involves dysfunction in such gene networks.

The present evidence of abnormal cell cycle networks involved in early brain maldevelopment in ASD is consistent within a larger and well-established animal model literature that cell cycle molecular machinery governs the overall size of the brain (Nakayama *et al*, 1996; Groszer *et al*, 2001; Chenn & Walsh, 2002; Ferguson *et al*, 2002; Feng & Walsh, 2004; Dehay & Kennedy, 2007; Ellegood *et al*, 2014) and supports the original theory proposed to explain early brain overgrowth in ASD (Courchesne *et al*, 2001). Here, we used the cell cycle network in leukocytes as an entry point to dissect the transcriptional alterations underlying pathology in living ASD infants and toddlers across developmental brain sizes, from small to abnormally large. Cell cycle network abnormalities involved substantial alteration of activity patterns of cell cycle hub-genes and modulation in expression of different peripheral genes. Such functional abnormality in the cell cycle network was greatest in ASD infants and toddlers who had bigger brain size, and moderate in those who had smaller brain size.

Abundant research shows that, beyond its effects on global brain and cerebral cortical size, cell cycle dysregulation can impact the core foundational framework of the fetal brain. Disruption may involve area-specific rates of cell production, cortical areal expansion, cell-fate determination, cell migration and differentiation, laminar specification, DNA integrity, the genesis of the connectivity of critical transient structures (subplate), and the generation of cytoarchitectonic maps (Galderisi *et al*, 2003; Dehay & Kennedy, 2007). Thus, we hypothesize disruptions in cell cycle regulation in early prenatal life may be a key defect underlying ASD. Such disruption may explain why some cortical regions have excess cells (Courchesne *et al*, 2011b) while other regions have too few (van Kooten *et al*, 2008). It may also explain why there are focal patches of disorganized prefrontal and temporal cortex in which cell-type and laminar-specific ISH expression are abnormally reduced (Stoner *et al*, 2014); this cortical pathology points to failure of the full normal cell- and laminar-fate program, which should be completed during the second and third trimesters (Stoner *et al*, 2014). More broadly, because the cell cycle disruption we identified appears to vary across affected ASD individuals with greater disruption associated with brain enlargement and lesser with smaller brain size, variation in the timing, nature and cellular location of its disruption could explain some aspects of variation in brain microstructural and functional outcome as well as clinical symptom heterogeneity in ASD.

Using a reverse genetic approach, we demonstrated that brain maldevelopment in ASD is likely due to the disruption of cell cycle networks, which in turn is related to key genes that have been frequently found mutated in ASD (De Rubeis *et al*, 2014). For the majority of high-risk ASD genes, the specific functional role and modalities of interactions are currently unclear. A recent literature review focusing on regulatory roles for genes in neurogenesis, neural induction, and neuroblast differentiation found that the vast majority of high-risk ASD genes help to regulate neural induction and early neuroblast development (Casanova & Casanova, 2014). Most importantly, the majority of core set genes influence neuronal development through multiple stages and are not limited to one single process. This pleiotropy of functions likely suggests that different modalities of interaction may co-exist, such as DNA-, RNA-, and protein-binding and may vary depending upon cell type and stage of development. The integrated Hc network displays a high content of regulatory elements strongly enriched in cell cycle phase-transition functions, suggesting that cell cycle length and timing may be one possible disrupted mechanism that can, at least partially, explain the downstream alteration of hub-genes associated with brain maldevelopment in ASD. This hypothesis is consistent with recent findings describing alterations of cell cycle timing and excess cell proliferation of neuroprogenitor cells derived from fibroblasts of living ASD patients who displayed early brain overgrowth (Marchetto *et al*, unpublished data). Several genes in the Hc network (e.g. *E2F1*, Cyclins, *MYC, CHD8, PIK3, AKT1, GSK3, PCNA, ERBB2*, Beta-catenin, and *SMAD3*) are known to play a role in G1/S and G2/M checkpoints, and to regulate neurogenesis (Zhu *et al*, 2003; Zhou & Luo, 2013). Some of these genes are also dysregulated in dorsolateral pre-frontal cortex of young ASD cases (e.g. *GSK3, AKT1*, Beta-catenin, *CREB1, SP1, TP53, SRC, FPS*, and *NODAL*; Voineagu *et al*, 2011; Chow *et al*, 2012). At the pathway level, PTEN, NOTCH and ESR transduction signals point to the same cell cycle phase-transition dysregulation. PTEN is a well-known tumor suppressor gene that counteracts the activation of PIK3/AKT in proliferation/self-renewal of neural progenitor cells, both *in vivo* and *in vitro* (Worby & Dixon, 2014). *In vivo*, its ablation leads to enhanced self-renewal capacity, accelerated G0-G1 cell cycle entry (Groszer *et al*, 2001, 2006; Gregorian *et al*, 2009), and faster transition from the G2/M to the G1 phase in embryonic stem cells resulting in overall shortened cell cycle (Kandel *et al*, 2002). Importantly, PTEN mutations have been found associated with enlarged brain size in ASD subjects (O'Roak *et al*, 2012) and one ASD PTEN brain has > 100% excess prefrontal neurons (Courchesne *et al*, 2011b; Courchesne personal communication). A target of PTEN signaling is the NOTCH pathway that as well represents a key regulator of neural stem cells (NSCs) maintenance. NOTCH inhibition has been shown to delay G1/S phase transition and commit NSCs to neurogenesis (Borghese *et al*, 2010). Similarly, the ESR1 pathway acts like a ligand-dependent transcription factor and promotes G1/S transition through several pathways (Prall *et al*, 1997; Foster *et al*, 2001).

Lastly, in addition to cell cycle dysfunction, we also found dysregulation of cell adhesion and protein folding networks in ASD toddlers. Alteration of cell adhesion functions, as in our case mediated by integrins, may have pleiotropic effects during both early and later developmental stages. During early cortical development neurogenesis, neuronal migration and cell specification are most active (Schmid & Anton, 2003), while later in development synapse formation, finalization and function become of primary importance (Milner & Campbell, 2002). Converging evidence shows that accumulation of misfolded proteins leads to the Unfolded Protein Response (UPR) (Walter & Ron, 2011) which in turn may

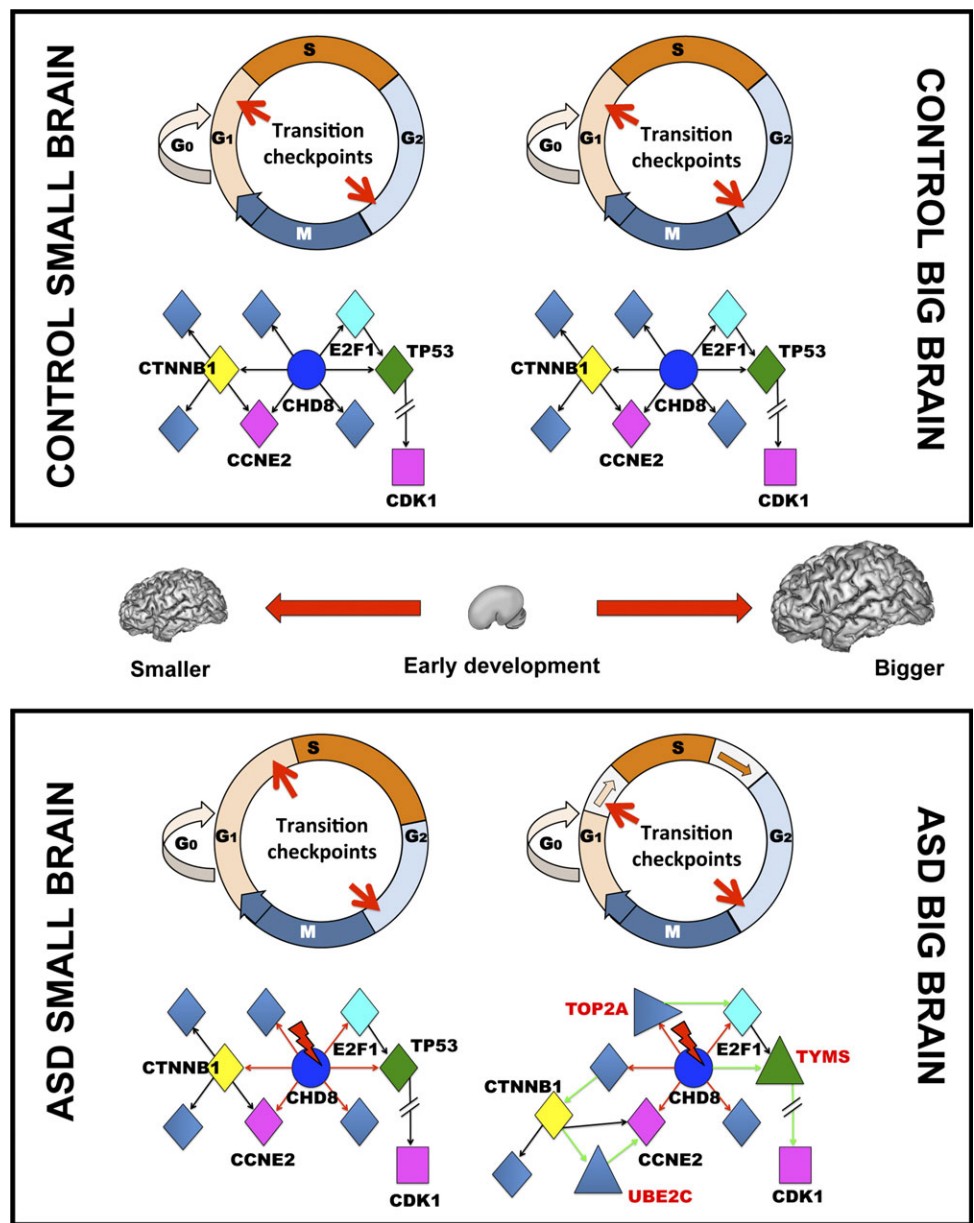

**Figure 7. Schematic representations of mechanisms that may underlie brain maldevelopment in ASD.**
In control brains (top panel), both smaller and bigger brains develop and function normally. At the cellular level, there are no significant alterations of cell division phases and the number of cells produced is within normal variation. At the molecular level, there are no genetic alterations and changes in gene expression are within normal variation with no significant alterations of network structure and function. In ASD brains (bottom panel), brain development is abnormal and smaller and bigger brains represent two different anatomical and functional outcomes. In smaller ASD brains, the cellular characteristics are currently less clear compared to bigger brains. We hypothesize that G1/S phase transitions may be longer and/or checkpoints may stall/delay the timing of cell divisions leading to a reduced number of cells. Alternatively, there are an increased number of apoptotic cells. These cellular phenotypes may be related to genetic mutations (red lightning bolt) of Hc genes that, for instance, regulate chromatin modification as in case of CHD8 (blue). The mutation leads to altered regulation (red arrows) of downstream transcription factors or regulatory elements (yellow, cyan, green diamond shapes) that in turn regulate the expression of brain-relevant genes (purple square). Mutated CHD8 can also alter directly the expression of brain-relevant genes, as in case of CCNE2. Gene expression and network functions are altered, but closer to normal brains. In bigger brains, cellular and molecular phenotypes are more pronounced compared to smaller ASD brains. Cellular evidence suggests that the increased number of neurons may be due to the shortening of G1/S phases. At the molecular level, this may be related to mutations and changes in gene expression that lead to a reorganization of networks controlling neuroprogenitor cell divisions. In addition to the downstream effects of Hc mutations (i.e. CHD8, red arrows), pronounced gene expression changes cause a substantial reorganization of the network with the activation of new regulatory genes (i.e., TOP2A, TYMS, triangle shapes) and new interactions (green arrows), thus leading to altered or different network functions.

underlie impaired synaptic function in ASD (Fujita *et al*, 2010; Falivelli *et al*, 2012), as well as global alterations of transcriptional regulation (Mendillo *et al*, 2012). Analysis of CHD8 knockdown

*in vitro* has provided evidence converging its downstream effects on transcriptional regulation to pathways we found altered in ASD (translation, cell cycle, protein folding, and cell adhesion).

   

## Conclusion

Our study provides compelling evidence to demonstrate that analyses of gene expression in peripheral blood allow the identification of functional genetic correlates of brain maldevelopment in ASD. Results point to convergent pathways of altered expression of genes and networks that ultimately lead to abnormal regulation of neuron production via defective G1/S phase transition during early stages of brain development (Fig 7), thus likely affecting neuroprogenitor divisions. While larger ASD brains display greater networks alterations with shorter G1/S phase timing (Marchetto *et al*, unpublished data) (Fig 7), the molecular underpinning of smaller brains remains less clear. It is plausible to hypothesize that either a lengthening of the G1/S phases or a checkpoint delay/malfunction may lead to reduced number of neurons or to increased cell removal (Fig 7).

Together with previous evidence (Courchesne *et al*, 2011b; Chow *et al*, 2012; Parikshak *et al*, 2013; Willsey *et al*, 2013; Stoner *et al*, 2014), we argue that disruption of gene networks related to specific prenatal genetic programs may underlie abnormal neuronal development in both small and large brains and may be part of the cause of ASD in a majority of individuals. Furthermore, genetic programs that control synapse development in early life provide a core functional neuronal component essential to further brain development and complex process elaboration; genetic defects in such genes have long been theorized, but not demonstrated, to impact postnatal brain growth and function in ASD (Parikshak *et al*, 2013). Further elucidation of these functional genomic pathologies underlying early brain development in ASD will facilitate research into biological targets for biotherapeutic intervention and development of accurate biomarkers for the early detection of babies and infants at risk for ASD.

# Materials and Methods

### Subject recruitment, tracking, and developmental evaluation

Research procedures were approved by the Institutional Review Board of the University of California, San Diego (No. 110049). Information on subject recruitment, evaluation, RNA samples, and gene expression procedures are provided also in a recent companion study (Pramparo *et al*, 2015). Toddlers were recruited via the 1-Year Well-Baby Check-Up Approach from community pediatric clinics (Pierce *et al*, 2011) as well as via referral from other community sources and evaluated using a battery of standardized and experimental tests that included: the Autism Diagnostic Observation Schedule (ADOS; Lord *et al*, 2000; Luyster *et al*, 2009), the Mullen Scales of Early Learning (Mullen, 1995) and the Vineland Adaptive Behavior Scales (Sparrow *et al*, 2005). Diagnoses were determined via these assessments and the Diagnostic and Statistical Manual, Fourth Edition (DSM IV-TR) (American Psychiatric Association, 2000). Testing sessions generally lasted 4 h and occurred across 2 separate days and the blood sample was usually taken at the end of the first day. Parents were interviewed with the Vineland Adaptive Behavior Scales (Sparrow *et al*, 2005) and a medical history interview. All toddlers were developmentally evaluated by a Ph.D. level psychologist and those that were younger than 3 years at the time

of blood draw were tracked every 6 months until their 3rd birthday when a final diagnosis was given. Only toddlers with a provisional or confirmed ASD diagnosis were included in this study. Our recent study (Pierce *et al*, 2011), which included the participation of 137 pediatricians who implemented > 10,000 CSBS screens showed that 75% of toddlers that fail the screen at the 1st year exam have a true delay (either ASD, language delay, global developmental delay or other condition). While ASD toddlers were as young as 12 months at the time of blood sampling, all but 3 toddlers have been tracked and diagnosed using the ADOS toddler module (Luyster *et al*) until at least age two years, an age where diagnosis of ASD is relatively stable (Cox *et al*, 1999; Kleinman *et al*, 2008; Chawarska *et al*, 2009). Toddlers received the ADOS module that was most appropriate for their age and intellectual capacity. Of the 87 enrolled ASD subjects, 64% had an ADOST, 31% had an ADOS 1, and 5% had an ADOS 2. Only toddlers with a provisional or confirmed ASD diagnosis were included in this study. Twenty-four final diagnoses for participants older than 30 months were also confirmed with the Autism Diagnostic Interview–Revised (Luyster *et al*). In order to monitor health status, the temperature of each toddler was taken using an ear digital thermometer immediately preceding the blood draw. If temperature was higher than 99, then the blood draw was rescheduled for a different day. Parents were also asked questions regarding their child's health status such as the presence of a cold or flu, and if any illnesses were present or suspected, the blood draw was rescheduled for a different day. The control group was comprised of typically developing toddlers as well as contrast toddlers (Table 1).

### RNA extraction, quality control and samples preparation

Four to six milliliters of blood was collected into EDTA-coated tubes from toddlers on visits when they had no fever, cold, flu, infections or other illnesses, or use of medications for illnesses 72 h prior blood draw. Blood samples were passed over a LeukoLOCK™ filter (Ambion, Austin, TX, USA) to capture and stabilize leukocytes and immediately placed in a −20°C freezer. Total RNA was extracted following standard procedures and manufacturer's instructions (Ambion, Austin, TX, USA). LeukoLOCK disks were freed from RNA-later and Tri-reagent was used to flush out the captured lymphocyte and lyse the cells. RNA was subsequently precipitated with ethanol and purified though washing and cartridge-based steps. The quality of mRNA samples was quantified by the RNA Integrity Number (RIN), values of 7.0 or greater were considered acceptable (Schroeder *et al*, 2006), and all processed RNA samples passed RIN quality control. Quantification of RNA was performed using Nanodrop (Thermo Scientific, Wilmington, DE, USA). Samples were prep in 96-well plates at the concentration of 25 ng/μl.

### Gene expression and data processing

RNA was assayed at Scripps Genomic Medicine (La Jolla, CA, USA) for labeling, hybridization, and scanning using expression BeadChips pipeline (Illumina, San Diego, CA, USA) per the manufacturer's instruction. All arrays were scanned with the Illumina BeadArray Reader® and read into Illumina GenomeStudio® software (version 1.1.1). Raw data was exported from Illumina GenomeStudio®, and data pre-processing was performed using

**Table 1. Summary of subject characteristics and clinical information.**

| Subject characteristics (all male) | ASD | control |
|---|---|---|
| AD | 77 | |
| PDD-NOS | 10 | |
| TD | | 41 |
| Other* | | 14 |
| Age in years – mean (SD) | 2.3 (0.7) | 2.7 (0.7) |
| Mullen scales of early learning (T-scores) – mean (SD) | | |
| Visual reception | 39.7 (11.0) | 53.55 (9.65) |
| Fine motor | 37.3 (12.2) | 55.85 (8.75) |
| Receptive language | 29.1 (12.0) | 49.65 (98.4) |
| Expressive language | 29.1 (11.4) | 50 (8.7) |
| Early learning composite | 71.0 (16.2) | 104.6 (11.9) |
| Autism diagnostic observation schedule (ADOS)[†] mean (SD) | | |
| ADOS CoSo/SA score | 15.0 (3.9) | 3.1 (2.7) |
| ADOS RRB score | 4.1 (1.9) | 0.4 (0.8) |
| ADOS total score | 19.1 (4.7) | 3.6 (3.2) |
| Vineland scores (VABS)[‡] | 82.2 (9.4) | 97 (8.45) |

* > Toddlers from "Other" category included language delay ($n = 9$), radiological abnormality ($n = 1$), premature birth but tests within the normal range on standardized tests ($n = 2$), socially emotionally delayed ($n = 1$), and prenatal drug exposure ($n = 1$).
[†]All toddlers received either the Toddler Module or Module 1 or 2, depending on age and verbal ability at time of testing. Sample: 64% of ASD population had ADOS T, 31% had ADOS Mod. 1, and 5% had ADOS Mod. 2.
[‡]Adaptive Behavioral Scales Adaptive Behavior Composite Score.

the *lumi* package (Du *et al*, 2008) for R (http://www.R-project.org) and Bioconductor (http://www.bioconductor.org; Gentleman *et al*, 2004).

Several quality criteria were used to exclude low-quality arrays as previously described (Chow *et al*, 2011, 2012). In brief, low-quality arrays were those with poor signal intensity (raw intensity box plots and average signal > 2 standard deviations below the mean), deviant pairwise correlation plots, deviant cumulative distribution function plots, deviant multi-dimensional scaling plots, or poor hierarchical clustering (Oldham *et al*, 2008). Five samples (four ASD and one control) were identified as low quality due to poor detection rates, different distributions, and curved dot plots, and were removed prior normalization. Eighteen (18) samples had 1 replicate and all pairwise plots of each replica had a correlation coefficient of 0.99. Hierarchical clustering of these replicated samples showed 13 samples having with the two replicas that clustered together; therefore, the B array was arbitrarily chosen for the following steps. For the remaining 5 of these replicated samples, the two replicas did not cluster together; thus, the averaged gene expression levels were used in the following steps. No batch effects were identified. BrB-array filtering Tool was used to obtain a final set of genes without missing expression values. Filtering criteria were Log Intensity Variation ($P > 0.05$) and percent missing (> 50% of subjects). A total of 142 final samples/arrays (87 ASD, 55 control), and thus 142 unique subject datasets, were deemed high quality and entered the expression analysis. Inter-array correlation (IAC) was 0.983. From among these subjects, 65 ASD and 38 controls had parental consent for MRI testing and neuroanatomical analysis.

## MRI scanning and neuroanatomic measurement

Scanning was performed with a 1.5 T GE scanner. A T1-weighted IR-FSPGR sagittal protocol (TE = 2.8 ms, TR = 6.5 ms, flip angle = 12 deg, bandwidth = 31.25 kHz, FOV = 24X cm, slice thickness = 1.2 mm, 165 images) was collected during natural sleep(Eyler *et al*, 2012). FSL's linear registration tool (FLIRT) rigidly registered brain images to a custom template that was previously registered into MNI space (Jenkinson & Smith, 2001). Registered images were then processed through FSL's brain extraction tool (BET) removing skull and non-brain tissue (Smith, 2002). Remaining non-brain tissue was removed by an anatomist to ensure accurate surface measurement. Gray matter, white matter, and CSF were segmented via a modified version of the FAST algorithm (Zhang *et al*, 2001) using partial volumes rather than neighboring voxels to increase sensitivity for detecting thin white matter in the developing brain (Altaye *et al*, 2008). The brain was divided into cerebral hemispheres, cerebellar hemispheres, and brainstem via Adaptive Disconnection (Zhao *et al*, 2010). Each cerebral hemisphere mask was subtracted from a sulcal mask generated by BrainVisa and recombined with the original FSL segmentation to remove all sulcal CSF voxels. The final hemisphere mask was reconstructed into a smoothed, three-dimensional mesh in BrainVisa to obtain surface measures (Rivière *et al*, 2009). ASD neuropathology exhibits larger brain size in a substantial proportion of affected toddlers (Courchesne *et al*, 2007, 2011a). In order to overcome trivial effects due to brain size differences between ASD and control toddlers, we randomly equilibrated the control sample with large-brain subjects. Total brain volume (TBV) measures were age-corrected using a generalized additive model (GAM-R package v1.06.2; Hastie & Tibshirani, 1995).

A semi-automated pipeline integrating features of FSL (www.fmrib.ox.ac.uk/fsl/) and BrainVisa (brainvisa.info) provided total brain volume (TBV). TBV measures were age-corrected using a generalized additive model (GAM-R package v1.06.2) (Hastie & Tibshirani, 1995).

## Weighted gene network co-expression and preservation analyses

Weighted gene network co-expression analysis (WGCNA) was used to identify gene modules across all 142 subjects and to calculate the first principal component of each module, herein called module eigengene (ME). All subjects were used in WGCNA to represent the largest gene expression variance available in each ME. Co-expression analysis was run by selecting the lowest power for which the scale-free topology fit index reached 0.90 and by constructing an unsigned (i.e., bidirectional) network with a hybrid dynamic branch cutting method to assign individual genes to modules (Pramparo *et al*, 2011). To generate random co-expressed modules for statistical analysis (see below), we scrambled 10,000 times module-color assignment for each gene and generated random co-expressed modules with the same number of genes of the real analysis.

Module preservation analysis (Langfelder *et al*, 2011) was used to test network structure against random chance in the combined WGCNA and to test comparability of modules structure between ASD and control datasets. Module preservation analysis was also used to compare network structure between the combined dataset and BrainSpan dataset. We assessed quality measures for the

Zsummary component with the related log.P-values ($-\log10$(P-values)) extracted from the preservation function output. Documentation for these types of analysis can be found here: http://labs.genetics.ucla.edu/horvath/CoexpressionNetwork/ModulePreservation/Tutorials/.

Two network metrics available in the WGCNA package were used to quantify networks perturbation and included: Gene Significance (GS) and Gene Connectivity (GC). GS is the correlation strength of gene expression levels with TBV and represents a measure of "biological significance or relevance" to a trait (Langfelder & Horvath, 2008). GC is a measure of correlation among co-expressed genes within a module and defines the gene-to-gene relationship, thus the level of gene connectivity (high/low) (Langfelder & Horvath, 2008; Langfelder *et al*, 2013). Genes with high GS values (ranked by high to low values) were considered highly relevant to brain size and genes with high GC values (ranked by high to low values) were considered centrally positioned in the co-expression network, therefore called hub-genes. The default minimum number of genes to constitute a module was 30. This threshold was used to define the top 30 genes that ranked highest using each network feature (GS and GC), thus providing information on their central role as relevance to brain (GS) and position within the network (GC). For instance, high connectivity (hub-genes) was defined by the highest 30 GC values of a module, while genes not ranking in the top 30 were deemed of low-connectivity (peripheral) genes.

### Statistical analyses

In our primary analysis, MEs were used in Pearson correlation tests to identify associations between modules and TBV across all available subjects. Adjusted P-values (q-values) were calculated across all 22 modules with the function *q*value() in the WGCNA package with default settings (lambda = 0,0.90,0.05 and method = "smoother"). Subsequent correlation tests were run within each diagnostic group and investigations focused only on modules with modest to high correlation coefficients ($r > 0.3$, $P < 0.05$ and FDR < 0.05). Bootstrapping on the correlation tests (10,000 resamples) was run to determine 95% confidence intervals around each correlation estimate. Permutation tests (10,000 iterations) were run to determine whether the association between MEs and TBV measures were specific to our dataset or given by chance. To do so, we scrambled 10,000 times module-color assignment for each gene and recalculated new module eigengene values (MEs) on each iteration. We then computed a correlation between ME and TBV for each of the 10,000 iterations. Finally, we calculated the P-value as the number of times within the null distribution, found a correlation as large or larger than the true correlation, and divided this by 10,001.

Linear modeling of MEs and TBV measures were followed by ANOVA analyses to test potential age-related effects on MEs significantly associated with TBV measures.

Permutation tests (10,000 iterations) were also run to test significant difference in correlation strength between ASD and control groups in ME-TBV correlations. To achieve this, we used a Fisher's z transform on the correlation values and computed the difference score between z-values (zDiff) for each group. Then within the permutation test, we shuffled group labels randomly, computed correlations, converted to Fisher's z, computed zDiff, and then iterated this entire procedure 10,000 times (each time with a different random shuffling of the group labels). To compute a P-value, we examined in the null distribution of zDiff values how often were values as large or larger than the zDiff value computed on the real (unpermuted) data.

Gene significance P-values for each gene represented in the networks were calculated using a Student asymptotic test for correlation within the WGCNA package. Quantification of difference between groups in GS-GC correlations was achieved by computing a z-statistic for the difference using the *paired.r* function from the *psych* R library. No P-values accompany this statistic because the statistic reflects the population estimate, given that all genes within the module are measured. Any z-statistic > 0 here reflects a difference between groups in GS-GC correlation strength.

Hypergeometric tests were run to establish significant enrichments against random chance in the Hc network and we provide the P-value as Hyp. P. The background pool total used in these calculations was 21,405.

Differentially expressed genes (DE; $P < 0.05$) was performed as previously described (Pramparo *et al*, 2015) to identify differentially expressed (DE) genes using a standard univariate two-sample *t*-test model with 10,000 random permutations in BRB-Array Tools.

### Functional enrichment, Hc network and PPI analyses

Pathway enrichment analysis was performed using the Metacore GeneGo platform, which provides a more extensive hand-curated, up-to-date gene annotation than available in other freely accessible sources (Scheiber *et al*, 2009; Shmelkov *et al*, 2011). Only enrichments with P-values (P) and FDR < 0.05 have been reported in this study. Metacore GeneGo was also used to generate a High-confidence (Hc) network using 33 reported Hc genes (De Rubeis *et al*, 2014). A total of 32 out of the 33 Hc genes were present in the GeneGO database and we used them as seeds to build a network by selecting the "create network" function, together with "downstream" and "by one interaction" options. The resulting Hc network included 414 genes. This Hc network therefore included subnetworks in which the 32 Hc genes were the main nodes and other genes were their direct targets. The Hc network was then color-coded based on other information.

We used the same strategy to identify to build upstream networks of the identified 23 candidate genes of brain maldevelopment. We used these 23 genes as seeds and created a network expanding upstream by one interaction. The list of upstream regulatory genes we identified included 106 genes. These were transcription factors and gene expression regulators. Subnetworks analysis followed the same methods used to test significant associations between MEs and TBV, but here we used the average value of expression for a subnetwork. Permutation tests (10,000 times) analogous to those run for testing the between-group difference in ME-TBV correlation strength were also ran here to test between-group difference in correlation strength (avg expression value—TBV measures).

DAPPLE software (http://www.broadinstitute.org/mpg/dapple/dappleTMP.php) was used for the genetic interaction and protein–protein interaction analysis using 1,000 permutations and a common interactor binding degree of 3.

## Data availability

Raw and normalized gene expression data are available at http://www.ncbi.nlm.nih.gov/geo/ with accession number GSE42133.

**Expanded View** for this article is available online.

## Acknowledgements

This research was supported by P50-MH081755 (EC), R01-MH080134 (KP) and R01-MH036840 (EC), KL2TR00099 and 1KL2TR001444 from UCSD CTRI (TP), NIH 1U54RR025204-01 (SM), and generous support from the Novo Nordisk Foundation that had been provided to the Center for Biosustainability at the Technical University of Denmark (NL). We would like to thank all families for making this study possible. We thank Roxana Hazin at UCSD ACE for her help with subjects recruitment; Marvin Javier at UCSD for the GAM analysis; a special thanks to Ondrej Libiger and Mary Winn at STSI and Christopher Woelk at UCSD, for their generous support and suggestions with the gene expression data analysis. We thank Peter Langfelder for his support with the WGCNA. We thank John Metz, Chris Willis and Sean McCreery from the Thomson Reuters team for the support with the Metacore GeneGo database. We thank William Brandler at UCSD for compiling and sharing an updated list of WES/WGS mutations, which helped us in the review process.

## Author contributions

TP and MVL performed data analysis, interpretation of results, and writing of the manuscript. TP and EC designed the study. KP and EC supervised the study and interpretation of data. KP oversaw the recruitment and psychometric testing of all of the subjects. KP, EC, and NL participated in writing of the manuscript. EC, KP, KC, SS, JY, MM, SM, and AD participated in MRI data collection and neuroanatomic measurement. KC developed the MRI processing stream. SSM participated in microarray processing. CAB participated in MRI data collection and samples management. CCB performed subject's assessment and determined diagnoses. LL performed blood collection. All authors read and approved the final manuscript.

## Conflict of interest

The authors declare that they have no conflict of interest.

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
