## [Review Process File · Molecular Systems Biology]

Cell Cycle Networks link Gene Expression Dysregulation, Mutation and Brain Maldevelopment in Autistic Toddlers

Tiziano Pramparo, Michael V. Lombardo, Kathleen Campbell, Cynthia Carter Barnes, Steven Marinero, Stephanie Solso, Julia Young, Maisi Mayo, Anders Dale, Clelia Ahrens-Barbeau, Sarah S. Murray, Linda Lopez, Nathan Lewis, Karen Pierce and Eric Courchesne

Corresponding author: Eric Courchesne, UCSD, Autism Center

Review timeline:

Submission date:	16 February 2015
Editorial Decision:	30 March 2015
Revision received:	15 June 2015
Editorial Decision:	05 August 2015
Revision received:	31 August 2015
Editorial Decision:	15 October 2015
Revision received:	26 October 2015
Accepted:	03 November 2015

Editor: Maria Polychronidou

Transaction Report:

1st Editorial Decision

30 March 2015

Thank you again for submitting your work to Molecular Systems Biology. We have now heard back from the two referees who agreed to evaluate your manuscript. As you will see from the reports below, the referees acknowledge that the presented findings seem potentially interesting. However, they raise a series of concerns, which should be carefully addressed in a revision of the manuscript.

Without repeating all the comments listed below, some of the more fundamental issues are the following:

- Statistical support needs to be provided for the main findings/conclusions reported in the study.
 - Further technical details and more clear descriptions of how the presented analyses were performed need to be included, in order to allow a more informed evaluation by the referees.
- Both referees provide detailed suggestions and comments regarding these two points.

If you feel you can satisfactorily deal with these points and those listed by the referees, you may wish to submit a revised version of your manuscript. Please attach a covering letter giving details of the way in which you have handled each of the points raised by the referees. A revised manuscript will be once again subject to review and you probably understand that we can give you no guarantee at this stage that the eventual outcome will be favorable.

 REFEREE COMMENTS

Reviewer #1:

I think that the paper is interesting but needs more statistical improvement.

This study studied gene expression in the leukocytes of ASD cases and controls, and identify gene modules associated with brain size development through their co-expression network analysis. This is an interesting and potentially important observation.

Major comments:

(1). The authors identified gene co-expression modules across different individuals in leukocytes. The functional implications of these co-expression modules are different from those identified in the earlier work, where co-expression modules were identified from different brain developmental stages (Parikshak, NN. et al., and Willsey, AJ. et al. 2013, Cell) or from different brain sections (Voineagu et al. Nature 2011, BA9/44). Intuitively, in the latter case, it is natural that genes showing co-expression across different time points or brain regions are functionally relevant, and thus partitioning genes into co-expression modules based on their "temporal" or "spatial" expression dynamics should reveal the true functional structure.

This study uses a different and novel approach in which the co-expression analysis identifies genes with similar expression patterns across different individuals.

The key idea of the manuscript is to correlate the average gene expression in each module across different individuals with the individuals' brain size, and several modules were identified for significant correlations. One important control is lacking in which, the authors need to perform permutation tests to determine whether or not these modules were expected by chance. Briefly, a set of degree-preserving shuffling for the co-expression network should be performed followed by the module detection procedure (with the identical parameters). For the permuted co-expression modules, what are the chances that they will show expression correlation with the brain size?

Specific comments:

(1). The statistical analyses in the manuscript are weak, and many places are just descriptive without justifying whether the observation was significant. For example, Fig. 2D, the author should use a box-plot to show the distribution of expression of all the genes in the module at each time point, rather than simply use the "eigengene value", which is the averaged expression without reflecting error bars. In the main text for this comparison, all the statements were purely descriptive based on the average expression. The authors need to examine whether or not these conclusions were statistically significant. In the same line, in later sections (the 3rd paragraph of the section of "Gene Correlation with TBV... ", all of the numbers were presented without statistical evaluation. It was the case for Fig. 5 - do these overlapping genes show statistical significance, or are just expected by chance? All these should be carefully examined before reaching a conclusion. There are many other parts in the text with similar problems, and the authors should carefully go through the manuscript, and provide more information to support their conclusions. Also, please give the exact P-values or numbers from the statistic tests, rather than $P < 0.05$ or $r > 0.2$, for example.

(2). Along the same line, many technical details were not presented, and thus many parts of the paper are hard to follow and the technical soundness cannot be judged. For example, in the last section for CHD8-analysis, they need to show FDR values and the statistical procedures of the enrichment tests. For the Sugathan PNAS paper, the CHD8 knockdown was performed in neural progenitor cells from a control patient (the GM8330-8 line), rather than from ASD subjects as stated in the main text. Also from the PNAS paper, it was shown that ~2000 genes were differentially expressed and more than 5000 genes were bound by CHD8, and these CHD8-regulated genes were enriched for neuronal functions - the authors should carefully compare the enrichment, and determine if the overlaps are indeed relevant to ASD etiology or for other non-specific functional categories given the blood expression data used in this study.

(3). In the co-expression analysis, the authors identified several modules, which showed difference in their correlation with brain size between case and control groups (Fig. 2C). The authors need to perform permutation tests to determine whether or not the difference is truly significant between case and control sub-groups. In a simple way, the case/control labels should be shuffled for, say, 10000 times, and determine how many times in the differential correlation could be observed the randomization.

(4). Differential expression analysis - how did you define differential expression (DE) ie. which threshold and what statistical procedures? The authors claimed that because the DE genes did not show significant overlap with cell cycle/protein-folding genes, they concluded that "Thus, the changes in expression we captured in our network-based approach were specific to variation of brain size for the cell cycle, protein folding and for about 50% of the cell adhesion genes." To me, the former does not imply the latter.

(5). Comparisons for GS/GC/MM: in the main text, the author first should explain what these abbreviations stand for, which makes the paper very hard to follow. Second, the three quantities are mutually dependent and highly correlated, making the comparisons redundant.

(6). The authors then examined the cell-cycle module using GeneMania, and reported their number of genetic interactions. As mentioned before, the numbers should be statistically compared. More importantly, what was the motivation to study human genetic interactions, how these are connected with ASD etiology, and what are the functional implications of a gene having or having not genetic interactions in this study. It is known that the epistatic interactions are pervasive in the human genome, but the human genetic interaction data are very scanty, so examining human genetic interactions cannot really add too much value in this comparison.

(7). The descriptions of the PPI network analysis and transcriptional regulation are rather jumbled, jumping from one to the other, and it is unclear what the authors want to say. Overall, statistics is weak in these sections: instead of giving descriptive text for the number of genes, the authors really need to perform statistic test to show if the number of differences was significant. Using the sentence "A few of these genes were also found associated to smaller and bigger brains in ASD..." is really weak and unconvincing.

(8). In the same section, many places are really hard to follow. How the Hc genes were derived and how many were tested in this study - how many overlapped with gene modules identified in this study? The authors hypothesized that "these Hc genes may be at least partially involved in the upstream regulation of the 23 PPI candidate genes" - what was the rationale behind this hypothesis? It has been known that many ASD genes are synaptic, and why do you expect them as transcriptional regulators of the 23 genes? Importantly, one gene SYGAP1 was analyzed as a regulator in this analysis, but it even does not have a DNA-binding domain. Under this hypothesis, why did the authors only choose to consider 32 ASD genes to construct a regulatory network? And how were the downstream targets identified/predicted from the Metacore, and in which cell type or tissue type they were identified? In a later section, for an over-representation test, the author compared the upstream regulatory genes and Hc network, and where did the numbers $n=106$ and 414 come from? In the earlier text, the authors said that they used 32 Hc genes for the analysis, but why $n=106$ and 414 here? All the missing information has made the paper very confusing.

Overall, the study has a number of interesting observations but the statistical tests should be strengthened and more technical details should be provided in the text.

Reviewer #2:

Pramparo et al. reanalyzed the data presented in Pramparo et al. (JAMA Psychiatry, 2015) from a different angle. Instead of analyzing differentially expressed genes in blood between ASD and control toddlers, the authors focus on gene correlated with total brain volume and identified cell cycle dysregulation associated with brain development in ASD. The results are interesting. However, I have many concerns on how they derived their results and conclusions.

1. on page 5, if the authors hypothesized and concluded gene networks were different between ASD and control toddlers and the differences were associated with brain size, why ASD and control data were put together to construct a coexpression network?
2. on page 6, I don't understand how correlation coefficients and associated p-values were calculated? Given 38 data points in the control set, $r=0.3$ corresponds to $p=0.067$ (how $p<0.001$ was calculated?). Similarly, there were 65 data points in the AST set, $r=0.2$ corresponds to $p=0.11$ (how $p<0.05$ was calculated?).
3. on page 7, what is "network activity"? whether module topology parameters can reflect "network activity" is questionable.
4. on page 9, it says "hub-genes relevant to TBV in ASD and control toddlers". What networks were used? The coexpression network based on the combined data, or ASD and control samples separately? It is very vague how "active" and "inactive" were defined at the end of the page. One fundamental question is whether there is an association between GS score and gene activity. Any in vitro assay can suggest the connection?
5. on page 11, it is not clear how CHD8-subnetwork was found. The author states that "we found that the strongest association" Without any statistics.

1st Revision - authors' response

15 June 2015

RESPONSE TO Reviewer #1:

I think that the paper is interesting but needs more statistical improvement.

This study studied gene expression in the leukocytes of ASD cases and controls, and identify gene modules associated with brain size development through their co-expression network analysis. This is an interesting and potentially important observation.

We thank the reviewer for considering our study interesting and important.

Major comments:

(1). The authors identified gene co-expression modules across different individuals in leukocytes. The functional implications of these co-expression modules are different from those identified in the earlier work, where co-expression modules were identified from different brain developmental stages (Parikhshak, NN. et al., and Willsey, AJ. et al. 2013, Cell) or from different brain sections (Voineagu et al. Nature 2011, BA9/44). Intuitively, in the latter case, it is natural that genes showing co-expression across different time points or brain regions are functionally relevant, and thus partitioning genes into co-expression modules based on their "temporal" or "spatial" expression dynamics should reveal the true functional structure. This study uses a different and novel approach in which the co-expression analysis identifies genes with similar expression patterns across different individuals. The key idea of the manuscript is to correlate the average gene expression in each module across different individuals with the individuals' brain size, and several modules were identified for significant correlations. One important control is lacking in which, the authors need to perform permutation tests to determine whether or not these modules were expected by chance. Briefly, a set of degree-preserving shuffling for the co-expression network should be performed followed by the module detection procedure (with the identical parameters). For the permuted co-expression modules, what are the chances that they will show expression correlation with the brain size?

The reviewer highlights two points here that we can address. The first regards a permutation-based test to ensure that modules that were detected are not detected by chance. We have now implemented an analysis that addresses this. In consultation with Dr. Peter Langfelder, who together with Dr. Horvath, developed the WGCNA package and tools, Dr. Langfelder kindly suggested that the degree-preserving shuffling suggested by the reviewer can be achieved through running a module preservation analysis where the two input datasets are the same dataset. This analysis will run permutations and assess quality measures such as the Zsummary component with the related

log₁₀ p values (-log₁₀(p-values)) that allow us to ascertain whether the modules were detected at greater than chance levels.

Results from this preservation analysis show that ALL modules have high scores for preservation and quality and they showed statistical significance against random chance. We have updated the methods and results sections accordingly and provided these results in supplementary material (see Figure E2).

The second point the reviewer makes here is about whether randomly generated modules would correlate with brain size. We addressed this point by scrambling 10000 times module-color assignment for each gene and re-calculating new module eigengene values (MEs) on each iteration. We then compute a correlation between ME and TBV for each of the 10,000 iterations. Finally, we calculated the p-value as the number of times within the null distribution we found a correlation as large or larger than the true correlation, and divided this by 10,001. We demonstrate that in the combined analysis (i.e. TD and ASD collapsed), apart for the yellow module, all other associations were significant. The same strategy was used on specific modules, but where we computed the correlation separately for each group (i.e. TD: greenyellow and grey60; ASD: salmon, turquoise, and cyan). These associations were also significant. We have now updated the results and methods sections accordingly and provided these analyses in supplementary material (see Figure E4 and E5).

Specific comments:

(1). The statistical analyses in the manuscript are weak, and many places are just descriptive without justifying whether *the observation was significant*. For example, Fig. 2D, the author should use a box-plot to show the distribution of expression of all the genes in the module at each time point, rather than simply use the "eigengene value", which is the averaged expression without reflecting error bars. In the main text for this comparison, all the statements were purely descriptive based on the average expression. The authors need to examine whether or not these conclusions were statistically significant.

First, we'd like to apologize for having provided an inaccurate definition in the manuscript of module eigengene (ME). A ME value is simply the first principal component scores from the expression values of all genes within a module. As such, the ME is a summary value of gene expression in the module. Because WGCNA already clusters genes into modules based on highly correlated expression, the ME as the first principal component captures the main axis of variation that is common to all the genes in that particular module and thus, reflects a good summary value for all genes within a module.

Regarding visual depiction of the results in Fig 2D of the previous submission, we would have to respectfully disagree with the suggestion to depict each individual gene as a boxplot. The reason here being that some modules have several hundreds of genes within them and several hundreds of boxplots would severely clutter up the visualization of the data and hence not be the preferred way to show the full variation of these modules across time. An alternative, which would be more appropriate, would be to use a scatterplot, as then each individual can be represented as a dot and the best fit regression line or curve can serve to be a visual indicator of the trend over time.

To more appropriately depict the data for this analysis, we have gone back and re-analyzed the BrainSpan dataset. In doing so, we have run a WGCNA on the BrainSpan data itself and then used module preservation analysis to indicate which of the modules detected in blood is highly preserved in the human brain across development. Here we find that the greenyellow cell cycle module that we heavily focus on is the most highly preserved of the blood modules. This module is also associated with normal TBV measures and, based on several pieces of evidence in the literature, is hypothesized to regulate brain development and size. This evidence from module preservation provides further independent verification of the greenyellow module's relevance to processes occurring in the human brain across development.

Next, to depict the developmental timecourse of this cell cycle module in BrainSpan, we have used a scatterplot to depict ME values over time, with each individual's ME value plotted as a dot (see Figure 3). To statistically verify that this cell cycle module is indeed upregulated during fetal compared to postnatal developmental timepoints, we have run an Wilcoxon rank sum test comparing

ME values for fetal timepoints versus postnatal timepoints. Here we find that fetal timepoints are upregulated relative to postnatal timepoints (Wilcoxon rank sum $z = 3.51$, $p = 4.44e-4$). These new analyses bolster our hypothesis that brain size maldevelopment in ASD is related to alteration of processes regulating cell number and provide a stronger basis to further investigate the network perturbation in the cell cycle module we describe next in the manuscript. We have updated the results and methods sections accordingly.

In the same line, in later sections (the 3rd paragraph of the section of "Gene Correlation with TBV...", all of the numbers were presented without statistical evaluation.

We have now provided more statistical description through the results, particularly in areas where the reviewer has pinpointed were lacking in clear evaluation or where the statistics were provided in supplementary material.

First, we have now used a permutation test to test whether groups differed with respect to the strength of the ME-TBV correlation. To achieve this we used a Fisher's z transform on the correlation values and computed the difference score between z-values (zDiff) for each group. Then within the permutation test, we shuffled group labels randomly, computed correlations, converted to Fisher's z, computed zDiff, and then iterated this entire procedure 10,000 times (each time with a different random shuffling of the group labels). To compute a p-value we examined in the null distribution of zDiff values how often were values as large or larger than the zDiff value computed on the real (unpermuted) data. These results are now provided in the supplementary material (see Table E3).

Similarly, we tested the correlation at the gene-level of two metrics (GC and GS) to assess associations with brain size. We questioned whether changes in Gene Connectivity (GC), which provides information on network organization, would correlate with changes in Gene Significance (GS), which is the correlation between gene expression and a trait (i.e. TBV), and whether the correlation would differ significantly between ASD and control groups. Fisher's z transform on the correlation values were significant for all three modules (greenyellow: $z = 3.47$; grey60: $z = 3.19$; salmon: $z = 2.48$). Since in this analysis we have measured the entire population of interest (all genes within each module) a p-value is not appropriate to compute. Instead, to assess the null hypothesis of $z = 0$, we can see from each module here, z is not equal to 0 in the population.

We moved these linear correlation scatterplots from the old Figure 3 to the new Figure 2 (see Figure 2Dii-Fii), and updated the results accordingly.

This analysis showed that within the three modules there is a significant change in network organization relevant to GS values between ASD and control. Our interpretation is that while the network organization from low to highly connected genes may be driving normal brain size in control, this relationship is significantly different in ASD.

Further insights into this network reorganization are provided by the analyses of the top 30 genes (ranked by GS and GC scores separately). First, we observed that the top 30 genes ranked by GS were different (by gene name) between control and ASD. Then we looked at the gene connectivity (GC) and observed that while in controls most of the correlation strength resides with the hub-genes, in ASD on the contrary resides in the peripheral genes. The findings suggest that there is a shift for brain relevance from hub- to peripheral genes in the cell cycle network of ASD subjects. These differences were all significant via statistical tests.

It was the case for Fig. 5 - do these overlapping genes show statistical significance, or are just expected by chance? All these should be carefully examined before reaching a conclusion.

To address reviewer's concern and provide statistical support for this data we have calculated GS p-values for each gene represented in these networks using a Student asymptotic test for correlation. This new analysis demonstrated that all genes described in this figure have significant p-values and thus their correlation with TBV is not due to chance. We now provide GS score and GS pValues for

each gene in the supplementary material (see supplementary File E1). The connectivity measure GC is strictly related to the network construction algorithm that WGCNA employs to generate the modules. With the preservation analysis (see above), we have demonstrated that these modules are not given by chance consequently measures of networks structure such as connectivity (GC), are as well specific to our dataset and their values have been now demonstrated to not be due to chance.

There are many other parts in the text with similar problems, and the authors should carefully go through the manuscript, and provide more information to support their conclusions. Also, please give the exact P-values or numbers from the statistic tests, rather than $P < 0.05$ or $r > 0.2$, for example.

We followed reviewer's suggestion and have indeed gone through an extensive revision of the manuscript to provide more information in support to our data. For instance, where the reviewer is referring to ($P < 0.05$ or $r > 0.2$), we now provide specific numbers (see Figure 2B). The tables we had in the supplementary material have been color-coded based on correlation strength and the specific p-values are provided in the main Figure 2B. Overall, in the main text we provide specifics for the statistical analyses.

(2). Along the same line, many technical details were not presented, and thus many parts of the paper are hard to follow and the technical soundness cannot be judged. For example, in the last section for CHD8-analysis, they need to show FDR values and the statistical procedures of the enrichment tests.

We agree with the reviewer that FDR values were not provided for the specific CHD8 enrichment analysis. We have updated in the methods that all the enrichment analyses have been run in Metacore GeneGO and enrichment with significant pValues and $FDR < 0.05$ are reported. We have revised the manuscript to improve technical soundness and provide missing technical details in results, methods or supplementary material.

For the Sugathan PNAS paper, the CHD8 knockdown was performed in neural progenitor cells from a control patient (the GM8330-8 line), rather than from ASD subjects as stated in the main text.

We thank the reviewer for spotting this incorrect description in the manuscript. We have corrected the improper sentences accordingly.

Also from the PNAS paper, it was shown that ~2000 genes were differentially expressed and more than 5000 genes were bound by CHD8, and these CHD8-regulated genes were enriched for neuronal functions - the authors should carefully compare the enrichment, and determine if the overlaps are indeed relevant to ASD etiology or for other non-specific functional categories given the blood expression data used in this study.

We thank again the reviewer for pointing us to the finding of more than 5000 CHD8 bound genes in the PNAS paper. This is valuable data that should be added to our study. Since three independent studies (Subtil-Rodriguez et al, 2014, Sugathan et al, 2014 and Cotney et al., 2015) are now available describing potential CHD8 targets, we decided to provide enrichment for all studies in the supplementary material. We also provide enrichment of their overlap, common CHD8 targets between Subtil-Rodriguez et al, 2014 and Sugathan et al, 2014 and common CHD8 targets between Sugathan et al, 2014 and Cotney et al., 2015. To further follow reviewer's suggestion we provide enrichment for the DE genes from the PNAS paper and the enrichment of the common genes found differentially expressed (DE) and target of CHD8. Surprisingly all the CHD8 targets enrichment were consistent with our previous findings showing translation genes, which we discovered in a recent blood-based signature of ASD (Pramparo, 2015), followed by cell cycle and protein folding genes, whose association with brain size is discussed in the current study. Analysis of the common DE genes and CHD8 targets from the PNAS paper displayed enrichment for signal transduction, transcription, development (WNT, beta-catenin, NOTCH, VEGF, IP3...) pathways, DNA-damage and cell cycle processes, all suggesting that knock-down of CHD8 may cause alteration in downstream processes associated with development. Lastly, analysis of the common CHD8 targets between Sugathan et al, 2014 and Cotney et al., 2015 displayed significant enrichments for transcription and cell cycle processes. In particular, cell cycle mitosis and S phase were top functions. These results can be found in supplementary material (see File E1).

(3). *In the co-expression analysis, the authors identified several modules, which showed difference in their correlation with brain size between case and control groups (Fig. 2C). The authors need to perform permutation tests to determine whether or not the difference is truly significant between case and control sub-groups. In a simple way, the case/control labels should be shuffled for, say, 10000 times, and determine how many times in the differential correlation could be observed the randomization.*

We have now used a permutation test to test whether groups differed with respect to the strength of the ME-TBV correlation. To achieve this we used a Fisher's z transform on the correlation values and computed the difference score between z-values (zDiff) for each group. Then within the permutation test, we shuffled group labels randomly, computed correlations, converted to Fisher's z, computed zDiff, and then iterated this entire procedure 10,000 times (each time with a different random shuffling). To compute a p-value we examined in the null distribution of zDiff values how often were values as large or more extreme than the zDiff value computed on the real (unpermuted) data. These results are now provided in the supplementary material (see Table E3).

(4). *Differential expression analysis - how did you define differential expression (DE) ie. which threshold and what statistical procedures? The authors claimed that because the DE genes did not show significant overlap with cell cycle/protein-folding genes, they concluded that "Thus, the changes in expression we captured in our network-based approach were specific to variation of brain size for the cell cycle, protein folding and for about 50% of the cell adhesion genes." To me, the former does not imply the latter.*

We had provided a description of the statistical procedure in the supplementary methods of this manuscript see "Differentially expressed genes (DE; $P < 0.05$) was performed to identify differentially expressed (DE) genes using a standard univariate two-sample t-test model with 10,000 random permutations in BRB-Array Tools. Significant threshold of univariate tests was 0.05" in Expanded view Methods. This method has been also published in a recent companion paper (Pramparo et al., JAMA Psychiatry 2015).

We do not fully understand what the reviewer means for "To me, the former does not imply the latter." however we'd like to address reviewer's concern by removing this marginal piece of data from our manuscript. Removing this information has no effect on the rest of the findings and does not alter in any way the take home messages of our study.

(5). *Comparisons for GS/GC/MM: in the main text, the author first should explain what these abbreviations stand for, which makes the paper very hard to follow. Second, the three quantities are mutually dependent and highly correlated, making the comparisons redundant.*

To avoid redundancy we provided explanations for these abbreviations and measures only the methods section, however we understand it may be helpful to have a description also in the results section. We have now added a description for these metrics in the results section.

To address reviewers concerns, we have removed the data from the module membership (MM) since it is highly correlated with the gene connectivity measure. We have retained in the manuscript the GC, which is fundamental to the description of the data related to the change in network organization and the GS feature, which is fundamental in the description of the genes that are most highly related to brain size variation.

(6). *The authors then examined the cell-cycle module using GeneMania, and reported their number of genetic interactions. As mentioned before, the numbers should be statistically compared. More importantly, what was the motivation to study human genetic interactions, how these are connected with ASD etiology, and what are the functional implications of a gene having or having not genetic interactions in this study. It is known that the epistatic interactions are pervasive in the human genome, but the human genetic interaction data are very scanty, so examining human genetic interactions cannot really add too much value in this comparison.*

The reason to look at the genetic interaction networks was to present independent support for our findings about the changes in network topology within the cell cycle module. We have demonstrated by looking at the gene expression changes that are relevant to variation of brain size in ASD vs control that a perturbation of the cell cycle network exists in ASD. This perturbation has led to a network re-organization in which hub-genes become less relevant to size (meaning their gene expression is weakly correlated to size) while peripheral genes acquire more importance in this synchronization between gene expression and brain size. It would be expected that such shuffling of genes would lead to a change in interactions of both PPI and genetic networks. However, we took into consideration reviewer's concerns about the lack of statistical measures to support the strength of these genetic interactions, and we have run statistical tests as described below. As reviewer anticipated these genetic interactions were not statistically significant. We thank the reviewer for pushing us to run this analysis. We have now eliminated this data from the manuscript and simplified the analysis and description of findings related to the network re-organization. Data description is now part of Figure 2.

Details on the GeneMania permutation analysis

We used Java-based command line tools for GeneMANIA (<http://pages.genemania.org/tools/>) to repeatedly run queries based on the real gene lists as well as repeated randomly selected gene lists from the background pool. To evaluate the gene interaction networks output by GeneMANIA, we computed network metrics such as number of edges, global efficiency, and the average clustering coefficient. For each permutation (10,000 iterations), we randomly selected from the background pool the same number of genes as the actual gene lists, re-ran a GeneMania genetic interaction query, and computed these same network metrics. To get the p-value, we counted up how many times in the permutation iterations, we obtained a value as large or larger than the values we obtained in the real data. Results for each network metric are shown in the table below, alongside the permutation p-value in parentheses.

	Number of Edges	Global Efficiency	Clustering Coeff
High Connectivity Control	39 (p = 0.24)	0.21 (p = 0.37)	0.10 (p = 0.14)
Low Connectivity ASD	46 (p = 0.32)	0.24 (p = 0.20)	0.04 (p = 0.47)

(7). *The descriptions of the PPI network analysis and transcriptional regulation are rather jumbled, jumping from one to the other, and it is unclear what the authors want to say. Overall, statistics is weak in these sections: instead of giving descriptive text for the number of genes, the authors really need to perform statistic test to show if the number of differences was significant. Using the sentence "A few of these genes were also found associated to smaller and bigger brains in ASD..." is really weak and unconvincing.*

With this revision of the manuscript we provide a much better description of the results and the statistical methods. Moreover we have performed an extensive revision of the results that are now supported by statistical analyses. To address the reviewer's concern about the DAPPLE PPI network we now provide in supplementary material: the PPI summary, the PPI network statistics, the list of direct connections and the seed scores. See supplementary File E1, tab "PPI analysis stats."

In regards to the sentence "A few of these genes were also found associated to smaller and bigger brains in ASD..." we believe it was misleading thus we have replaced it with "Two Hc genes were also found mutated in subjects with significantly smaller (DYRK1A) and bigger (CHD8) heads". Here we cited a study (O'Roak et al., Science, 2012) in which the authors report on two genes (DYRK1A and CHD8) that have been found mutated in subjects with significantly smaller and bigger heads. The reason for citing this reference is to provide the reader with some context taken from recent studies and to highlight the importance of integrating genetic findings of gene mutation with our functional genomics evidence. In particular, we describe the use of 32 out of 33 high confidence genes that have been found mutated in large cohorts of ASD subjects. Most importantly, mutations in these genes have been described in relationship with smaller and bigger brains, thus providing a functional and pathological role for these Hc genes. This is an important aspect for our study since we are trying to demonstrate that there is convergence of data between our evidence and genetic studies where for the most part are lacking of functional relationship to ASD pathophysiology. Therefore, we do not see how that sentence in particular is weak and unconvincing.

(8). *In the same section, many places are really hard to follow. How the Hc genes were derived and*

how many were tested in this study - how many overlapped with gene modules identified in this study?

We apologize with the reviewer for providing confusing information. We have revised the description of the methods and results in order to clarify all the points that have been flagged by the reviewer.

The Hc genes were derived from one of the most important studies in the genetics of ASD (De Rubeis et al, 2014). This study identified LoF mutations in a large cohort of ASD subjects providing the most accurate information (up to date) about genes that may be involved in the etiology of ASD. Some of these genes have been described to have genetic liability for ASD but also found mutated in subjects with significantly smaller or bigger brains in ASD (see O’Roak, Science 2012; CHD8, DYRK1A). We have used the strongest candidate list of 33 Hc genes, of which 32 were present in the Metacore database. These 32 were used for the construction of the Hc network.

The authors hypothesized that "these Hc genes may be at least partially involved in the upstream regulation of the 23 PPI candidate genes" - what was the rationale behind this hypothesis?

Our rationale is critical to our study and in the field of ASD since most of the genes that have been found mutated in ASD patients and are thought to be causing ASD, lack functional downstream information. For the most part we know that they are mutated in a very small proportion of ASD subjects but we do not know the functional consequences of their mutation and/or whether their mutation have any effects at the transcriptomic or proteomic level. Thus, here we present how the functional abnormalities we have found in the transcriptome of ASD toddlers is impressively only 1 step (interaction) away from genes that have been found mutated in larger studies where functional effects for these genes is unknown and whether their effect have anything to do with neuro-developmental processes that may be disrupted in ASD.

It has been known that many ASD genes are synaptic, and why do you expect them as transcriptional regulators of the 23 genes?

Unfortunately, this is an incomplete, biased and inaccurate description of the candidate genes in ASD. It is true that the first few genes that have been found mutated in ASD were synaptic genes. However, we now have many more candidate genes with different functional roles and it is well established that genetic mutations (both common and rare) do not affect exclusively synaptic functions. In terms of the importance of transcriptional regulator genes, there are many candidate ASD genes like this, with the best example being CHD8. Genes that regulate transcription was also an important point made by a recent exome sequencing paper that has highlighted more recurrent de novo loss of function variants (De Rubeis et al., 2014, Nature).

Importantly, one gene SYGAP1 was analyzed as a regulator in this analysis, but it even does not have a DNA-binding domain. Under this hypothesis, why did the authors only choose to consider 32 ASD genes to construct a regulatory network? And how were the downstream targets identified/predicted from the Metacore, and in which cell type or tissue type they were identified?

We purposely included all the Hc genes as identified and described in De Rubeis et al, 2014 study to not bias the selection only of genes with specific functions. Our analysis was to demonstrate a convergence of findings to a network with potentially pleiotropic functions and tried to understand or lay the hypothesis to later understand how these genes may interact and their roles expressed in ASD.

In a later section, for an over-representation test, the author compared the upstream regulatory genes and Hc network, and where did the numbers n=106 and 414 come from? In the earlier text, the authors said that they used 32 Hc genes for the analysis, but why n=106 and 414 here? All the missing information has made the paper very confusing.

We have improved the description of the methods and results sections to fill in the missing information and performed several additional statistical analyses that support our findings.

Overall, the study has a number of interesting observations but the statistical tests should be strengthened and more technical details should be provided in the text.

We thank the reviewer for the positive final comment. We hope we have addressed all concerns in a satisfactory way.

RESPONSE TO Reviewer #2:

Pramparo et al. reanalyzed the data presented in Pramparo et al. (JAMA Psychiatry, 2015) from a different angle. Instead of analyzing differentially expressed genes in blood between ASD and control toddlers, the authors focus on gene correlated with total brain volume and identified cell cycle dysregulation associated with brain development in ASD. The results are interesting. However, I have many concerns on how they derived their results and conclusions.

1. on page 5, if the authors hypothesized and concluded gene networks were different between ASD and control toddlers and the differences were associated with brain size, why ASD and control data were put together to construct a coexpression network?

The primary WGCNA analysis we have run is indeed on data combined across both groups. There is a practical yet important reason behind this. For our aims of examining the association between a metric such as the module eigengene to brain size, and then comparing differences between groups, the only way to achieve this is by running WGCNA on all data from both groups. Running separate WGCNA analyses, one per group, will not ensure that a module in one group will be exactly the same as the same module in another group (e.g., they may be comprised of different subsets of genes). Therefore, the only way to get a metric like the module eigengene, that is comparable across both groups, for the purposes of relating it to brain size, is to run one WGCNA analysis on all the data from both groups.

We have now run further analyses that verify that module structure is relatively well preserved across both groups, thus, justifying the decision to run WGCNA on all data from both groups combined. With module preservation analysis we now show that all modules detected in one group are highly preserved across the other group (see Figure E3). Therefore, there are not fundamental differences in network structure that make ASD and TD co-expression networks so different that they cannot be compared. Rather, the module preservation analysis indicates that network structure is highly preserved and thus similar. This allows us to compute summary measures like the module eigengene, for examination of association with brain size.

2. on page 6, I don't understand how correlation coefficients and associated p-values were calculated? Given 38 data points in the control set, $r=0.3$ corresponds to $p=0.067$ (how $p<0.001$ was calculated?). Similarly, there were 65 data points in the ASD set, $r=0.2$ corresponds to $p=0.11$ (how $p<0.05$ was calculated?).

To follow overall reviewer's suggestions we have much improved the statistical analyses and provided the related information in the main text. For this particular instance, we have color-coded the correlation coefficients and provided p-values in Figure 2B.

It is not clear to us what the reviewer is referring to, for instance with this comment " $r=0.3$ corresponds to $p=0.067$ ". We found a significant correlation ($P=0.0003$) with coefficient of 0.3 (see the Grey60 module in Fig. 2B) in the combined analysis, thus using all 103 subjects, not using only the 38 controls. Similarly, we found a significant correlation ($P=0.01$) with coefficient of 0.21 (see the Salmon module in Fig. 2B) using all 103 subjects and not only the 65 ASD subjects. In the separate group analysis we found significant correlations with coefficients >0.44 using 38 control subjects and >0.27 using 65 ASD subjects. In this section of the study we also provide a permuted analysis to support the non-random association between module eigengene and TBV measures (see Figure E4 and E5).

3. on page 7, what is "network activity"? whether module topology parameters can reflect "network activity" is questionable.

We agree with the reviewer that network activity may lead to over- or mis-interpretation of the data. We replaced the term "activity" with "pattern".

4. on page 9, it says "hub-genes relevant to TBV in ASD and control toddlers". What networks were

used? The coexpression network based on the combined data, or ASD and control samples separately? It is very vague how "active" and "inactive" were defined at the end of the page. One fundamental question is whether there is an association between GS score and gene activity. Any in vitro assay can suggest the connection?

We have clarified the sentence to specify that we are using hub-genes from the co-expression cell cycle module generated from the combined analysis. We have run separate WGCNA analysis in ASD and control groups only to validate the different modules that we found associated to TBV in the two groups from the combined analysis. All the other analyses were performed using the data from the combined WGCNA analysis since only in that case we can compare module. We cannot compare module structures and related information from the modules identified in the separate WGCNA analyses because they would not be entirely comparable and thus the data would be biased. We have addressed this point above providing additional information with the preservation analysis.

We agree with the reviewer that using the "active" and "inactive" terms may be vague in our context, thus we have modified the manuscript by replacing them with "correlated" and "uncorrelated".

Reviewer's suggestion to follow up on our findings with *in vitro* assays is a good idea and fit the purpose of our study which is to guide future analyses using model systems. However, for this study we have not planned to include in vitro analyses in the current design, but we are heading in that direction.

5. on page 11, it is not clear how CHD8-subnetwork was found. The author states that "we found that the strongest association" Without any statistics.

We have greatly improved the description of the subnetworks generation and analysis (see specific results section in main text). Of all the subnetworks, three were found with significant associations between the average expression value of the subnetwork and TBV measures. After permutation analysis only the CHD8 subnetwork showed correlation strength that were significantly different between ASD and controls after 10000 tests. This is a remarkable finding showing that one of the top candidate gene in autism is found to functionally affect gene expression related to brain size in toddlers with ASD.

2nd Editorial Decision

05 August 2015

Thank you again for submitting your work to Molecular Systems Biology. First of all, I would like to apologize for the exceptional delay in getting back to you. We have now heard back from the reviewer (#1) who agreed to evaluate your manuscript. As you will see below, this reviewer raises a number of remaining concerns, which we would ask you to address in a revision of this work.

On a more editorial level we would like to ask you to address the following issues: listed below.

- ~~–Please include a Data Availability section after the Materials and Methods, providing the Accession number for the newly generated expression data.~~
- ~~–Please include in the Materials and Methods the statement regarding the informed consent procedures (currently provided as a response to point 12 of the Author Checklist).~~
- ~~–The Expanded Methods are not particularly lengthy and can therefore be included in the main text.~~
- ~~–Please provide legends/descriptions for the data described in the file labeled as "File E1". These descriptions can be included in an additional tab in the .xls file or in a README .txt file provided together with the Dataset in a .zip file. File E1 should be renamed and referred to in the text as Dataset E1.~~

 REFEREE COMMENTS

Reviewer #1:

The authors have improved the readability of their manuscript and have also improved the statistical calculations where were largely omitted in the earlier version. There are still a few places requiring further clarification (including a few more statistical issues):

(1) The goal of this paper is to identify the genetic components underlying brain size development using gene expression data from leukocytes. However, it was confusing at the beginning of the paper (Intro and Results) why the authors did not group patients with enlarged or normal brain, and then compare with their respective matched controls to identify differentially expressed genes (responsible for brain size). It is unclear why the authors set out to build a co-expression network rather than perform a direct comparison. At least the authors should elaborate more on their motivation underlying these studies.

(2) The authors need to provide more information regarding the case/control samples - it is unknown from the main text whether these samples have been matched by age/sex/ethnicity, etc. In some individual places of the paper the authors mentioned 'age-correction', but I am not sure if age/ethnicity/sex factors have been considered throughout all the analysis. At the very beginning of the Result section, the authors should provide an overview of the data and the samples for all the relevant information.

(3) It is great that the authors have performed additional statistical analysis. In the manuscript, the authors showed gene co-expression pattern showed difference in ASD patients than in control subjects (Fig.2B and GS-GC correlations). However, the dataset are not balanced because there are more ASD individuals than control subjects - The authors need to comment on whether their analysis are sensitive to sample size.

(4) On. Pg. 5 "Seven modules" were significantly correlated with TBV...", since you are comparing the correlation across all the modules, the authors should correct the P-values and present the significance (Table 2B) using false discovery rates.

(5) I still have concern with the section where the authors tried to link the cell cycle network with genes from the previous literature.

a. Instead of making connection with the previously identified genes, the authors should download the published exome-seq/CNV data and go straight to examine whether the identified cell cycle network genes shown an increase in their mutation burden. This is an important step to connect the observation found here with previously published studies.

b. The authors stated that "To frame our transcriptional findings in the context of these genomic studies,..., we tested the hypothesis...". This section is really weak. First of all, throughout the paper, it is all about gene co-expression analysis, not involving any analyses of 'transcriptional factor binding/ upstream factor, etc', so why did the authors assume or hypothesize the cell-cycle genes are their downstream targets? In fact, many of the 32 ASD genes are synaptic genes, and how can they regulate cell cycle genes? I found this part is extremely weak.

c. The target-upstream relationship was from a software MetaCore GeneGO - how good are the annotation could achieve? Are these targets by PWM scan or by motif analysis, and how can these be compared with ENCODE targets?

d. In the late section, the authors used DE genes for comparison - this went back to my earlier question, why did not use these DE genes at the very beginning of the manuscript (Comment #1). And more relevant information should be given to these DE genes rather than just say DE genes followed with a citation - this is confusing as it is unclear whether the DE genes actually came from this study or from the cited work.

e. I would suggest leave out the predicted up-stream factor analysis, and only present the cell-cycle genes in the context of CHD8-mediated network. Current writing is simply descript, only showing the enrichment without further biology (even did not indicate FDRs), and the authors should examine the shRNA-knockdown data in NPCs/NSCs in Cotney et al. and Sugathan et al., and explore if the identified cell cycle genes shown overall perturbation upon CHD8 knockdown - this will further link your blood data with brain data. Perhaps leave out the data from Subti-Rodriguez, etval, which came from an irrelevant cell line.

RESPONSE TO Reviewer #1:

The authors have improved the readability of their manuscript and have also improved the statistical calculations where were largely omitted in the earlier version. There are still a few places requiring further clarification (including a few more statistical issues):

(1) The goal of this paper is to identify the genetic components underlying brain size development using gene expression data from leukocytes. However, it was confusing at the beginning of the paper (Intro and Results) why the authors did not group patients with enlarged or normal brain, and then compare with their respective matched controls to identify differentially expressed genes (responsible for brain size). It is unclear why the authors set out to build a co-expression network rather than perform a direct comparison. At least the authors should elaborate more on their motivation underlying these studies.

We apologize to the Reviewer that the introduction and approach to examining our question was not sufficiently clear. The approach we chose is because brain size in ASD at young ages is continuous and not dichotomous. Figure 1 in this Response shows this continuum of brain size in ASD, for postmortem brain weight from Courchesne et al. (JAMA, 2011) and for head size from O'Roak et al (Science, 2012). They and other such studies show that ASD involves an abnormal *shift* towards larger size, rather than a subtype of normal and another of enlarged. Thus, while ASD involves early brain overgrowth, the effect is a statistically significant *shift in distribution* towards larger size and weight. So, the neural and genomic mystery has been what underlies this shift in brain size distribution at young ages in ASD.

The approach we have taken in this paper is designed to address this. That is, our approach identifies associations between two continuous, quantitative variables of interest (brain size, gene expression) in ASD as compared to controls. In doing so, we have taken an analysis approach that is tailored to reveal systems biology organization and co-expression network analysis that illuminate the functional genomic bases of the brain size continuum at the ages of first detection in ASD.

This approach also achieves our aims in statistically principled way by removing redundancy (i.e. thousands of genes are clustered by similarity into discrete co-expression modules). Furthermore the way in which we have detected such associations involves preserving the quantitative and continuous nature of the variables.

Lastly, as discussed further in our response to Reviewer comment #5 below, many high confidence ASD genes regulate cell cycle function and proliferation, such as CHD8, and are not cell cycle genes per se. This suggests that effects on cell proliferation (and brains size) may be quantitative and continuous and not categorical (which might be predicted if ASD high-confidence genes were key hub cell cycle genes per se).

We have now inserted text into the introduction that explicitly highlights the aim of intending to take a system biology approach to understanding how variation in co-expression modules is associated with the continuous variation in brain size in ASD at young ages.

(2) The authors need to provide more information regarding the case/control samples - it is unknown from the main text whether these samples have been matched by age/sex/ethnicity, etc. In some individual places of the paper the authors mentioned 'age-correction', but I am not sure if age/ethnicity/sex factors have been considered throughout all the analysis. At the very beginning of the Result section, the authors should provide an overview of the data and the samples for all the relevant information.

We have updated the Results section with information about the dataset, which have been previously reported in Pramparo et al, 2015 (JAMA Psychiatry).

See first paragraph in section “Different Gene Networks Associate with Brain Size in ASD and Control Toddlers”.

All study subjects are males, and there is no sex bias. We have run Pearson’s Chi-squared and Multivariate regression analyses to test whether differences in race and ethnicity between ASD and control subject could bias the results, and found no significant association. Differences in age have been taken into account throughout the analyses. Thus, as described in the previous revision of this manuscript, we used a generalized additive model (GAM) to age-correct brain total volumes using the control brains as normative measures (see Methods). Moreover, we tested by linear modeling and ANOVA analysis that the association between age-corrected brain volumes and module eigengenes (for greenyellow, grey60 and salmon) remained significant after age-correction (see Results).

We now also include the age-correction analysis by linear modeling and ANOVA testing for the association between CHD8-subnetwork gene expression and TBV that, as expected, was independent of age. We have added this result in Appendix Tables (see Table S7) and updated the results accordingly. These several analytic steps show that our main results are not confounded by age differences between ASD and control.

(3) It is great that the authors have performed additional statistical analysis. In the manuscript, the authors showed gene co-expression pattern showed difference in ASD patients than in control subjects (Fig.2B and GS-GC correlations). However, the dataset are not balanced because there are more ASD individuals than control subjects - The authors need to comment on whether their analysis are sensitive to sample size.

We thank the Reviewer for asking for clarification. Yes, it will be the case that sample size does have some impact on sample statistics (e.g., Pearson correlations), in terms of telling us how precise the sample statistic is of the honing in on the statistic in the population. Larger sample sizes give more precise sample statistic estimates (i.e. smaller confidence intervals) than smaller sample sizes. Therefore, the sample statistic in the ASD group will likely be a more precise estimate of the population statistic. We have now run bootstrapping (10,000 resamples) to derive 95% confidence intervals around the sample statistics for each group. These confidence intervals are now included. With respect to tests of between-group differences, the sample sizes of each group are included in computations for these tests, and thus they are to some degree being accounted for when determining such statistics.

(4) On. Pg. 5 "Seven modules" were significantly correlated with TBV...", since you are comparing the correlation across all the modules, the authors should correct the P-values and present the significance (Table 2B) using false discovery rates.

The Reviewer is correct that correlations are computed here across all modules. We have calculated the adjusted p-value (q-value) across all modules using the WGCNA function `qvalue()` and verified that all 7 modules pass $FDR < 0.05$. We have updated the legend to Figure 2B, the Results and the Methods sections accordingly.

(5) I still have concern with the section where the authors tried to link the cell cycle network with genes from the previous literature.

a. Instead of making connection with the previously identified genes, the authors should download the published exome-seq/CNV data and go straight to examine whether the identified cell cycle network genes shown an increase in their mutation burden. This is an important step to connect the observation found here with previously published studies.

We thank the Reviewer for suggesting alternative experiments to push forward the study of the genes we identified that are relevant to brain size in ASD. We are indeed working in this direction with a larger study that focuses on rare and common variation associated with different clinical features, including brain size; however this effort has only recently started. In response to comment 5.b, we provide further explanation for making a connection to previously identified genes.

In the current study we are not aiming to identify and propose new candidate genes for loss of function analysis but rather to add functional information to a well-defined set of candidate genes whose roles has not been fully studied in the context of ASD pathophysiology. It is important to emphasize that most genetic studies in ASD have not established functional relevance to clinical aspects of ASD, such as brain enlargement, language responsiveness, treatment outcome and so forth. It is reasonable to hypothesize that by using a multimodality approach, as in our case between gene expression levels and neuroanatomical measures, it will be possible to identify genes and networks that are disrupted in ASD and responsible for the phenotypes. Moreover, this approach is better suited to reveal ASD signatures that are specific to subgroups of ASD subjects. This is unlikely to be the case if we focus our investigations to only previously-identified genetic alterations for which we have plenty of evidence demonstrating the rarity of each event and the low penetrance and expressivity.

In addition, our approach does not rely on a specific type of mechanism for gene dysregulation such as the case of loss of function mutations. It is very likely that other types of mechanisms may affect gene expression regulation, for instance via variants that affect regulatory elements, via short non-coding RNA or via epigenetic changes. Thus, the analysis of gene expression dysregulation that takes into account multiple possibilities is an effective way to understand the biological processes that during both prenatal and postnatal development underlie the ASD phenotypes. Our study is a successful outcome in this direction.

To directly follow the Reviewer's suggestions, we analyzed an up-to-date list of identified mutations, both non-synonymous (non-syn) and loss-of-function (LoF), from the following WES/WGS studies: O'Roak et al., (2011), O'Roak et al., (Science 2012), O'Roak et al., (Nature 2012), Sanders et al., (2012), Iossifov et al., (2012), Iossifov et al., (2014), Jiang et al., (2013), Neale et al., (2012), De Rubeis et al., (2014) and Yuen et al., (2015). We tested the difference in mutation burden for the 253 gene-set present in the cell cycle module between ASD vs control subjects. For both non-syn mutations and LoF we found no statistical difference between the two groups:

Stats	non-syn	LoF
Odds ratio	1.0532	1.3798
95 % CI:	0.5173 to 2.1439	0.2783 to 6.8416
z statistic	0.143	0.394
Significance level	P = 0.8864	P = 0.6935

We also downloaded the latest list (June 2015) of CNVs from the SFARI database and analyzed the gene content of CNVs with size <2Mb to test for significant enrichment in the cell cycle genes we identified. Hypergeometric analysis demonstrated that the enrichment was not statistically significant ($P=0.951$).

Furthermore, as part of the larger study mentioned above, we recently completed the genotyping and CNV analysis of the same 142 subjects from this study. To satisfy reviewer's curiosity, we can anticipate that 122 subjects (73 ASD and 49 control) displayed CNVs identified using methods previously described in Chow et al, 2012 and Pramparo et al 2015. We ran ODDs ratio and fisher exact probability test to look for differential CNV burden in the cell cycle module genes, Hc network genes and predicted upstream regulators of the brain relevant genes. Although a few previously identified single ASD candidate genes were found involved by CNVs (for instance CTNND1 and AUTS2), we have not found a statistically significant difference between ASD and control subjects. This suggests that CNVs are not the primary mechanism for gene dysregulation in our dataset, thus confirming the hypothesis that other mechanisms not based on loss of function may alter the regulation of gene expression relevant to brain size.

Altogether, these negative results support our hypothesis-driven approach that upstream regulatory genes destabilize cell cycle functions and lead to abnormal cell proliferation and control of neuron production.

b. The authors stated that "To frame our transcriptional findings in the context of these genomic studies,..., we tested the hypothesis...". This section is really weak. First of all, throughout the paper, it is all about gene co-expression analysis, not involving any analyses of 'transcriptional factor binding/upstream factor, etc', so why did the authors assume or hypothesize the cell-cycle genes are their downstream targets? In fact, many of the 32 ASD genes are synaptic genes, and how can they regulate cell cycle genes? I found this part is extremely weak.

Several lines of evidence sustain our hypothesis. First, about 14 years ago, the senior author hypothesized (Courchesne et al., Neurology 2001; study that discovered early brain overgrowth on ASD) that dysregulation of cell cycle genes may cause excess cell proliferation and brain overgrowth; he subsequently demonstrated that ASD prefrontal cortex has a 67% excess of neurons (Courchesne et al., JAMA, 2011) and recently an ASD animal model demonstrates that excess neuron proliferation due to abnormal sustained proliferation is sufficient to lead to main ASD neural and behavioral defects (Fang et al., Cell Reports, 2014). Second, at the pathway level, genes with cell cycle functions have been found implicated in ASD (Pinto et al., AJHG, 2014; Pinto et al., Nature, 2010). Third, we have previously shown genetic association of cell cycle genes in ASD and differentially expressed cell cycle genes in young ASD cortices (Chow et al., 2012). As further described below, we show in a new table (see last Tab in Dataset E1), 16 of the 32 high confidence ASD genes affect cell number production, including dysregulation of cell cycle functions (see for instance CHD8, PTEN, DYRK1A, ANK2, ARID1B, NR3C2, POGZ).

More broadly, abnormal proliferation and brain growth are related to many high confidence ASD genes that are known to affect prenatal neuron proliferation. Most of these ASD gene defects are associated with brain enlargement and/or excess neuron number (e.g., CHD8, PTEN, EIF4A, WDFY3, KCTD13-CUL3-RhoA)(Yang et al., 2014; O'Roak et al., Science 2012; Orosco et al., 2014; Dong et al., 2009; Lin et al., 2015) and a few are associated with abnormally reduced brain size and cell proliferation in model systems (DYRK1A)(O'Roak et al., Science 2012, O'Roak et al., Nature 2012). WDFY3 was not previously known as a cell cycle gene, and has the following GO functional annotations with no obvious relationship to cell cycle functions: aggrephagy, autophagy, galactosyltransferase activity, catabolic process, ion binding, lipid binding, metal ion binding, response to stress. Nonetheless, loss of WDFY3 in mouse recapitulates several key aspects of the ASD brain pathology including brain enlargement and cortical organization and function. In particular, the WDFY3 animal model of autism displays increased proliferation, frontal cortical areal expansion, focal laminar disorganization and brain enlargement (Orosco et al., 2014).

Further, CHD8, a gene of high importance in ASD, strongly targets cell cycle and DNA damage networks (Cotney, et al., 2015; Sugathan et al., 2014). Cotney et al. (2015) write, "Genes that showed the strongest differential expression due to CHD8 knockdown (EdgeR Poisson P value $< 1.68 \times 10^{-6}$ and absolute \log_2 fold change > 0.1) were enriched in cell cycle functions, as well as transcriptional regulation, reinforcing the observations obtained from the pathway analysis". The cell cycle pathway included chromatin interacting proteins, remodelers and modifiers, and the P53 and HIPPO pathways influence WNT signaling. In ASD postmortem frontal cortex, we found significant dysregulation of cell cycle and WNT genes (Chow et al., 2012). These genes and pathways are in our leukocyte diagnostic classifier (Pramparo et al., 2015). Thus, the strongest dysregulated genes and pathways in neural models of CHD8 in ASD are also those dysregulated in leukocytes in ASD infants and toddlers, further indicating the value of information about regulatory functions in leukocytes at the early age of initial ASD detection, as we propose here.

Moreover, CHD8 is more strongly expressed in blood than in most brain regions (see Figure from bioGPS expression pattern at <http://biogps.org/>). CHD8 expression is as robust in prefrontal cortex as in blood: Prefrontal cortex is a key region of structural, functional, cellular and molecular disorder in ASD, including a large excess of neurons and focal laminar dysplasia (Courchesne et al., 2005; Courchesne et al., JAMA 2011; Courchesne, Brain Res. 2011; Stoner et al., 2014; Allely et al., 2014; Lainhart et al., 2015). Thus, leukocytes may be an especially valuable window to *in-vivo* CHD8-related genomic and phenotypic abnormality in ASD at young ages.

In sum, to further address the Reviewer's question about HC ASD genes, we added a new table with functional GO terms and descriptive information of these 32 HC ASD genes from the MetaCore database. The MetaCore database is available by subscription and has the unique advantage of being carefully manually curated and backed by validation experiments by experts in systems biology. Please see response to comment 5.c for additional information.

The new table added to the Dataset E1, demonstrates that among the 32 high confidence (HC) genes (annotated in Metacore) 16 genes have roles in the regulation of cell number (keywords: cell cycle, proliferation, apoptotic, cell death), 15 have roles in the regulation of other genes (keywords: transcription, chromatin, gene expression), 12 genes have roles in synaptogenesis, synapse function, axonal growth (keywords: synapse, synaptic, axon guidance, action potential, neuron projection) and 8 genes have other roles (keywords: cytoskeleton, transport, transduction). This evidence shows that: 1) the majority of HC genes have functional roles in the regulation of cell number and/or regulate expression of other downstream genes; 2) only 12 of the HC 32 genes have "synapse-specific" or synapse-only" functional roles; and 3) multiple genes have pleiotropic functions. For instance, ANK2 have roles in all three categories or PTEN has a role in cell proliferation and neuron-neuron synaptic transmission. Mutations in PTEN are also associated with enlarged brain.

This pleiotropy of gene functions may be explicated at different time points in development and it is part of the fundamental dogma in the study of ASD in which brain pathology is constantly changing from early developmental stages to early postnatal periods. Thus the functional genomic basis of ASD is far more complex than just disruption of synapse development and function.

In a recent review from Geschwind and State (2015; in particular see Table) we found several elements supporting our hypothesis and rationale: 1) genes found to carry multiple mutations are functionally heterogeneous. In addition to synaptic function, evidence exists for roles in a wide range of other cellular functions. These functions include the regulation of other genes; 2) de novo missense mutations appear also to show a risk signal in aggregate, of smaller magnitude than for LoF mutations. This suggest that a network analysis is more appropriate to study disrupted biological processes and that dysregulation of gene expression, rather than ablation of gene expression, may be a key mechanism; 3) The growing set of risk genes identified by statistical methods, without reference to an a priori biological hypothesis, has also enabled relatively unbiased assessments of the representation of pathways or processes relevant to ASD. This suggests that the study of ASD without a bias towards a specific gene function (such as synaptic) will be a more valuable approach to our understanding of ASD pathoetiology. 4) Recent studies by others as well as our lab have aimed to address when mutations might be having their effects, and results point to mid-fetal development. Our findings point to the same if not earlier time points of development. 5) Rare de novo variants affect early transcriptional regulators likely in most cells, while previously identified mutations (SFARI) are found in later-expressed genes of cortical cells. This suggests that more research is needed to understand the processes and mechanisms that are affected during early stages of development, for instance when the outer subventricular zone is expanding followed by cortical patterning.

We deleted the confusing sentence "To frame our transcriptional findings in the context of these genomic studies" in the results section.

c. The target-upstream relationship was from a software MetaCore GeneGO - how good are the annotation could achieve? Are these targets by PWM scan or by motif analysis, and how can these be compared with ENCODE targets?

We'd like to provide some information in regards to the quality of database Metacore GeneGO. In contrast to publicly available databases (such as DAVID) and/or other subscription-based pathway analysis tools, MetaBase (Metacore's database) content is manually curated by a team of over 150 PhD and MD employees of GeneGo (Thomson Reuters) who read full text articles and use small experiment information.

We encourage the reviewer to read the following two articles:

- 1) Title: "Assessing quality and completeness of human transcriptional regulatory pathways on a genome-wide scale." <http://www.ncbi.nlm.nih.gov/pubmed/21356087>. The NYU group compared 10 pathway databases. The results of their study show that for the majority

of pathway databases, the overlap between experimentally obtained target genes and targets reported in transcriptional regulatory pathway databases is surprisingly small and often is not statistically significant. The only exception is the MetaCore pathway database, which yields statistically significant intersection with experimental results in 84% of cases.

- 2) Title: “Gaining insight into off-target mediated effects of drug candidates with a comprehensive systems chemical biology analysis.” <http://www.ncbi.nlm.nih.gov/pubmed/19434832>. This paper speaks to the quality of MetaBase. This database covers metabolism, signaling, toxicity, disease states and the quality of information is very high and clean. This is important, as any analysis with these tools is based on this information, so a high quality database is a fundamental necessity. The publication from Novartis that states “MetaBase is, arguably, the most comprehensive manually curated database of mammalian biology and medicinal chemistry data available. Overall, it contains over 6 million experimental findings on protein-protein, protein-DNA and protein-compound interactions. MetaBase also includes thousands of established signaling and metabolic pathways, ligand-receptor information for known drugs, drug targets and diseases, kinetic information on drug-metabolizing enzymes and relevant signaling proteins, ontologies for diseases, functional processes, toxicities, proteins and drugs.”

Below we provide a reference as example of interaction

Qu L, Huang S, Baltzis D, Rivas-Estilla AM, Pluquet O, Hatzoglou M, Koumenis C, Taya Y, Yoshimura A, Koromilas AE
Endoplasmic reticulum stress induces p53 cytoplasmic localization and prevents p53-dependent apoptosis by a pathway involving glycogen synthase kinase-3beta.
 Genes & development 2004 Feb 1;18(3):261-77
 PMID: 14744935

Experiment Details

Note	p53 state	Mechanism	Effect	GSK3 beta state	Methods
p53 physically interacts with GSK3beta and increases its activity	P53_HUMAN	Binding	Activation	GSK3B_HUMAN	protein kinase assay, coimmunoprecipitation, in vitro

As far as other suggestions by the Reviewer, we don't fully understand the proposed comparison since it doesn't seem to be an “apples-to-apples” comparison. To our knowledge, PWM scans are powerful for predictive protein-DNA interactions, cannot account for protein-protein interactions. Similarly, ENCODE focuses on DNA sequence-based analysis.

In any case we fully understand where the Reviewer is coming from and it is a good suggestion to consider other databases in the future to further study the different aspects of these interactions.

d. In the late section, the authors used DE genes for comparison - this went back to my earlier question, why did not use these DE genes at the very beginning of the manuscript (Comment #1). And more relevant information should be given to these DE genes rather than just say DE genes followed with a citation - this is confusing as it is unclear whether the DE genes actually came from this study or from the cited work.

In response to comment #1 we have provided several points to support the use of our network strategy as compared to single-gene level analysis. The end goal was to leverage the interactive nature of genes and identify candidate networks associated with brain maldevelopment in ASD. This strategy is also supported by several points we made in response to comment 5.b.

Our study started from the analysis of multiple networks associated with variation in brain size. From multiple networks we purposely narrowed down our analysis to a much smaller set of interacting genes in a network that displayed the most severe disruption and with relevance to brain development. This allowed us to pinpoint disrupted mechanisms and single genes with the largest effects on the network and relevance to brain size. At this point of the analysis, it is relevant to show which genes in the Hc network may be disrupted at the systemic level. Moreover, we show that 8 out of the 32 Hc genes are differentially expressed in blood. This evidence further supports that the Hc network we constructed from Hc genes is key and demonstrates an unexpected convergence of findings with functional relevance to the pathophysiology of the ASD brain.

We clarified in the results section from which study the DE genes come from.

e. I would suggest leave out the predicted up-stream factor analysis, and only present the cell-cycle genes in the context of CHD8-mediated network. Current writing is simply descript, only showing the enrichment without further biology (even did not indicate FDRs), and the authors should examine the shRNA-knockdown data in NPCs/NSCs in Cotney et al. and Sugathan et al., and explore if the identified cell cycle genes shown overall perturbation upon CHD8 knockdown - this will further link your blood data with brain data. Perhaps leave out the data from Subti-Rodriguez, etval, which came from an irrelevant cell line.

We appreciate the suggestions presented here, and have addressed several of them as follow:

In response to the above comments, we have provided clear evidence in regards to the pleiotropy of roles of HC genes, which include the downstream regulation of other genes. The best example we provide is with CHD8, however in a new table, we have found that among the HC genes, 14 of 32 HC genes have transcriptional, chromatin modification or gene expression regulatory roles described in an up-to-date protein-protein database. Additional examples can also be found for genes that are not within the HC gene list. A2BP1 (also known as RBFOX1) has been widely described to regulate neuronal splicing networks clinically implicated in neurodevelopmental disease, including ASD (Voineagu et al., 2011; Fogel et al., 2012).

By following the reviewer's suggestion we also demonstrated that LoF or non-synonymous mutations or CNV are not likely the mechanism responsible for the gene expression dysregulation we identified in cell cycle genes associated to brain size variation in ASD. This new evidence supports our hypothesis that upstream regulatory genes destabilize cell cycle functions.

We have also provided important information to highlight the quality and reliability of the database (Metacore) that we used for the network interaction analysis.

In the methods section we had stated that we used an FDR threshold of 0.05 for all enrichments. In Dataset E1 we now have updated the enrichment table to which the reviewer is referring to (predicted upstream factors) with FDR values.

Lastly, we believe we have also provided several compelling elements to demonstrate that the HC network is an important point of convergence for gene mutation, gene expression regulation and genes relevant to brain development. We understand that the reviewer might see this section of our study as more exploratory; however we were able to show relevant evidence. For example, we found that 8 HC genes are predicted to regulate downstream genes that we found altered in ASD and relevant to brain size. Four of these genes are differentially expressed in blood, thus suggests a systemic gene expression alteration. These results are both supported by existing literature evidence, as to the case of CHD8 and are also novel. This will promote new hypothesis-based research for genes that so far have been lacking a functional role in the understanding of ASD pathoetiology. Thus, we think it will be valuable to keep the upstream factors information since it would create a gap in our study rationale and in the interpretation of the HC network

We followed the reviewer's suggestion and removed the data presented in Figure 6 and Dataset E1 related to the publication from Subti-Rodriguez et al.

We also tested the Reviewer's hypothesis to look for enrichment in the cell cycle genes we have identified with genes perturbed after CHD8 knockout in the two studies from Cotney et al. and Sugathan et al. To be more stringent we used only the common genes from the two studies and tested both the hNSC and human brain dysregulated gene lists (Cotney et al. 2015). As the reviewer predicted, we found statistically significant enrichment (hypergeometric $P=6.3e-05$) between the cell cycle genes we identified and the overlapping perturbed genes from Cotney hNSC and Sugathan. A non-significant enrichment (hypergeometric $P=0.9$) instead was found between the cell cycle genes and the overlapping genes from Cotney human brain and Sugathan. These findings have been now included in the results section.

We have modified Figure 6 by removing the enrichment data from Subti-Rodriguez et al and included the enrichment of the overlap between Cotney human brain and Sugathan, which was already presented in Dataset E1.

3rd Editorial Decision

15 October 2015

Thank you again for submitting your work to Molecular Systems Biology. We have now heard back from the referee who agreed to evaluate your manuscript. As you will see below, the referee is now satisfied with the modifications made and only lists two minor issues, which should be addressed in a revision of the work.

Moreover, we would ask you to address ~~the following~~ some editorial issues listed below:

~~Please include a short table of contents in the beginning of the Appendix.~~

~~Please make sure that you refer to Appendix Figures and Tables as "Appendix Figure SX" or "Appendix Table SX" both in the main text and the Appendix PDF.~~

~~We have slightly edited the synopsis text (attached below). Please let us know if would like to introduce further modifications.~~

REFeree COMMENTS

Reviewer #1:

The authors have substantially improved this manuscript, and most of my concerns have been nicely addressed-- I appreciate the authors' effort to make these comparisons solid.

Two minor points that the author should consider for further improvement:

(1) In my comment (5b), because the authors are stressing that it is the upstream factors that explain their observed gene dys-regulation, my question was how the 32 ASD genes identified by the authors could support their claim about "upstream factors"? In this revision, it is good that the authors now showed that 14 out of the 32 genes were regulatory - the authors should rephrase their statement (showing the overall enrichment perhaps), because the remaining (more than half of their identified 32 genes) appears to weaken their conclusion. As discussed in my earlier comments, some genes in this gene list do not have DNA-binding domains so their role as an upstream regulatory factor is not clear (or at least should be discussed)? As an example, the authors quoted RBFOX1 in their response letter to support this notion. However, RBFOX1 is an RNA-binding protein and does not have a DNA-binding domain. Although some RNA-binding proteins also bind DNA, this just at least be discussed.

(2) $OR=inf$ does not make sense, is this because of numerical overflow. The authors should change it.

RESPONSE TO Reviewer #1:

(1) In my comment (5b), because the authors are stressing that it is the upstream factors that explain their observed gene dys-regulation, my question was how the 32 ASD genes identified by the authors could support their claim about "upstream factors"? In this revision, it is good that the authors now showed that 14 out of the 32 genes were regulatory - the authors should rephrase their statement (showing the overall enrichment perhaps), because the remaining (more than half of their identified 32 genes) appears to weaken their conclusion. As discussed in my earlier comments, some genes in this gene list do not have DNA-binding domains so their role as an upstream regulatory factor is not clear (or at least should be discussed)? As an example, the authors quoted RBFOX1 in their response letter to support this notion. However, RBFOX1 is an RNA-binding protein and does not have a DNA-binding domain. Although some RNA-binding proteins also bind DNA, this should at least be discussed.

We agree with the Reviewer that the role of most of these 32 high confidence (Hc) genes is currently unclear. This is why there is the need for functional genomic studies such as this one to link the identification of genetic mutations to functional phenotypes. In our study the functional phenotypes are gene expression dysregulation and brain size variation. The reviewer is raising a valid point that some genes among the 32 ASD Hc list do not have DNA-binding domains. However, we similarly advocate that their functional role as regulators may well be through protein-protein interaction (PPI) and/or RNA-binding interaction. Indeed, the Hc-network that we propose as a potential point of convergence for gene dysregulation relevant to brain size variation is based not on DNA-binding but on PPI. Thus, much work has to be done in order to characterize the modality of interactions and the downstream effects caused by the interruption of these interactions.

In the last submission, we provided a table of GO terms (Dataset E1) for the 32 Hc genes to help the reviewer and the reader appreciate the pleiotropy of functions associated with these genes without necessarily focusing on one particular modality of interaction (DNA-binding, RNA-binding or PPI). Our intent was to highlight the more recent notion that ASD candidate genes are not necessarily only synaptic genes but involve also the regulation of other downstream genes. Moreover, we aimed to stress the point that an approach agnostic to an *a priori* functional hypothesis is more advantageous and enables unbiased assessments of the representation of pathways and processes relevant to ASD.

However, we do agree with the reviewer's suggestion, and we have now added a paragraph to the discussion about the unclear functional role of these genes in ASD.

"For the majority of high-risk ASD genes, the specific functional role and modalities of interactions are currently unclear. A recent literature review focusing on regulatory roles for genes in neurogenesis, neural induction and neuroblast differentiation, found that the vast majority of high-risk ASD genes help to regulate neural induction and early neuroblast development (Casanova & Casanova, 2014). Most importantly, the majority of core set genes influence neuronal development through multiple stages and are not limited to one single process. This pleiotropy of functions likely suggests that different modalities of interaction may co-exist, such as DNA-, RNA- and protein-binding and may vary depending upon cell type and stage of development."

Lastly, we also tested the hypothesis of whether the 32 HC genes represent (to a substantial degree above chance levels) genes involved in transcription regulation. Hypergeometric test displayed a p-value in for transcriptional regulation enrichment that is above random chance ($P=0.0015$).

(2) OR=inf does not make sense, is this because of numerical overflow. The authors should change it.

The Inf values arose previously due to the presence of a zero in our 2x2 table used to compute the odds ratios. In these instances, we follow the general practice recommended by Pagano & Gauvreau (Pagano & Gauvreau, 2000 Principles of biostatistics. 2nd ed. Belmont, CA: Brooks/Cole) to add

0.5 to all cells of the table before recomputing the odds ratios. This eliminates the Inf values. Accordingly, we have corrected the test values in the main text of the manuscript.